



# Role of *Calanus sinicus* (Copepoda, Calanoida) on dimethylsulfide production in Jiaozhou Bay

Juan Yu[1,2], Jiyuan Tian[3], Zhengyu Zhang[1], Guipeng Yang[1,2*], Hongju Chen[4]

[1]Key Laboratory of Marine Chemistry Theory and Technology, Ministry of Education; College of Chemistry and Chemical
Engineering, Ocean University of China, Qingdao, 266100, People's Republic of China
[2]Laboratory for Marine Ecology and Environmental Science, Qingdao National Laboratory for Marine Science and Technology, Qingdao 266071, People's Republic of China
[3]College of Food Science and Engineering, Qingdao Agricultural University, Qingdao, 266009, People's Republic of China
[4]College of Environmental Science and Engineering, Ocean University of China, Qingdao, 266100, People's Republic of
China

*Correspondence to*: Guipeng Yang (gpyang@ouc.edu.cn)

**Abstract.** The role of copepod *Calanus sinicus* on the production of dimethylsulfide (DMS)/dimethylsulfoniopropionate (DMSP) in Jiaozhou Bay was evaluated in field and laboratory experiments. Samples at 10 sites in the bay were collected monthly from June 2010 to May 2011 (except for March 2011), and zooplankton species composition was analyzed. The
relationship between copepod abundance and DMS or DMSP concentration was investigated. Effects of *C. sinicus* grazing on DMS/DMSP production at different conditions (i.e., algal diets, food concentrations, and salinities) were assessed in the laboratory. Data from the field experiment showed that *C. sinicus* was the predominant copepod in Jiaozhou Bay (up to 123 individuals m$^{-3}$ in May 2011) and has no apparent effect on DMS/DMSP production. In the laboratory experiment, compared with *Gymnodinium* sp. or *Emiliania huxleyi*, *C. sinicus* feeding on *Isochrysis galbana* and *Chaetoceros curvisetus*
exhibited increased DMS concentration, whereas high salinity inhibited DMS production. This study indicated that DMSP was transferred from phytoplankton to copepod body, fecal pellet, and seawater through copepod grazing. Our results provided important information to understand the biogeochemical cycle of DMSP in Jiaozhou Bay.

Key words: Copepod; Diet; Dimethylsulfide; Dimethylsulfoniopropionate; Grazing; Salinity

## 1 Introduction

Dimethylsulfide (DMS) is the most abundant biogenic sulfur gas that may influence planetary climate by forming cloud condensation nuclei that alters global radiation balance (Charlson et al., 1987). The biogeochemical cycling of dimethylsulfoniopropionate (DMSP), which is the main precursor of DMS, recently came under close scrutiny. The marine environment is the major source of DMSP, and DMSP is synthesized by many marine phytoplankton species as an osmolyte (Kirst, 1996). Therefore, the dynamics of DMSP in the ocean will have an important influence on global DMS production.
The conversion of DMSP to DMS is regulated by complex trophic processes in the water column, e.g., algal senescence (Nguyen et al., 1988), phytoplanktonic enzyme catalysis (Niki et al., 2000), bacterial activity (Kiene and Linn, 2000), and



zooplankton grazing (Dacey and Wakeham, 1986; Wolfe et al., 2000; Yu et al., 2015). Currently, most studies have focused on field research concerning the spatial and temporal distributions and fluxes of DMS/DMSP (Turner et al., 1996; Wong et al., 2005). Many field studies indicated a weak correlation between DMS and parameters directly related to primary producers (Kettle et al., 1999). Numerous abiotic and biotic factors, e.g., factors other than phytoplankton, might play an

important role in DMS and DMSP dynamics. Zooplankton are known to account for a large fraction (10% to 25%, or higher) of the daily oceanic primary production (Lancelot and Billen, 1985), and knowledge about the role of zooplankton grazing in the DMSP biogeochemical processes is scarce. Copepods, a trophodynamic link between primary and tertiary productions, play a key role in the cycling of materials and energy in marine ecosystems (Kiørboe, 1997). Dacey and Wakeham (1986) were the first to show that copepod grazing stimulated DMS production in a laboratory experiment. Other studies have also

documented that zooplankton grazing induced DMS production (Belviso et al., 1990; Christaki et al., 1996; Daly and DiTullio, 1996; Leck et al., 1990) because of sloppy feeding (Dacey and Wakeham, 1986; Tang et al., 2000) or enhanced DMSP lyase activities (Wolfe and Steinke, 1996). The DMSP ingested by zooplankton was accumulated in the body and transferred to the upper food chain, and a portion of the DMSP ingested was transferred to fecal pellets, which were subsequently uncoupled as DMS and DMSP productions.

The calanoid copepod *Calanus sinicus* Brodsky is distributed in the East China Sea, the Yellow Sea, and the coastal waters of Japan (Brodsky, 1965) and is one of the predominant zooplankton in the East China Sea, the Yellow Sea, and the Jiaozhou Bay. In this study, we performed field and laboratory experiments to investigate the effects of copepod on DMS and DMSP productions in Jiaozhou Bay. The field experiments revealed the relationship between DMS/DMSP and biotic or abiotic parameter, and the laboratory experiments indicated the changes of DMS, dissolved and particulate DMSP ($DMSP_d$

and $DMSP_p$), DMSP in copepod bodies ($DMSP_z$), and DMSP in fecal pellets ($DMSP_f$) under different conditions of salinities, food concentrations, and diet species. The data obtained in this study will further recognize the role of copepods in the DMSP biogeochemical cycle.

## 2 Materials and methods

### 2.1 In situ experiment

#### 2.1.1 Sampling site

Fieldwork was performed at 10 stations of the Jiaozhou Bay from June 2010 to May 2011 once a month, except of March 2011 due to strong wind (Fig. 1). The Jiaozhou Bay (36°7′24.44″ N, 120°14′44.3″ E), a semi-enclosed bay in Qingdao City (China), separates Huangdao District from Qingdao City and borders on Jiaozhou City. The bay is 32 km long and 27 km wide, with a surface area of 362 km².



### 2.1.2 Abundance and taxonomic composition of zooplankton

Zooplankton samples were collected by vertical tows using a conical–cylindrical plankton net with a 50 cm mouth opening and a 160 μm mesh size. A flow meter was used to estimate the amount of filtered water. We retrieved the net at 0.3 m s$^{-1}$ to 0.5 m s$^{-1}$ at each station. Samples were then rinsed into cod-end buckets, concentrated, and preserved in 5% formalin for qualitative and quantitative analyses of in situ zooplankton. Taxonomy was determined by optical microscopy. In situ abundance was calculated using tow volume estimates, which were determined by net dimensions and flow meter values.

### 2.1.3 Abiotic parameters and analyses of chlorophyll a (Chl a) and bacteria

The shipboard measurements of temperature and salinity of surface water were obtained with the Sea-Bird 911 plus a conductivity–temperature–depth probe. Water samples were filtered using Waterman GF/F filters (nominal pore size of 0.7 μm) to determine Chl $a$. Chl $a$ concentrations were measured using a Hitachi F4500 fluorometer according to the methods of Parsons et al. (1984). Based on the procedures of Porter and Feig (1980), bacteria were counted by epifluorescence microscopy (Hitachi F4500; total magnification of ×1,000).

## 2.2 Incubation experiment

### 2.2.1 Phytoplankton and zooplankton cultures

Phytoplankton species (*Isochrysis galbana*, *Chaetoceros curvisetus*, *Emiliania huxleyi*, and *Gymnodinium* sp.) were provided by the Marine Microalgae Research Center, Ocean University of China in different sizes and DMSP contents. They are widespread species in Jiaozhou Bay and Chinese coastal waters (Li et al., 2005; Zhong et al., 2001). These algae are described in Table 1, and the sizes of algal cells are obtained by measuring at least 200 cells using a calibrated ocular micrometer with a light microscope at a magnification of × 400. Cell volume was estimated by approximating the shape of each species to an elliptic sphere, which subsequently determined the biovolume according to the method of Verity et al. (1992). In this study, we determined the cellular carbon contents in phytoplankton according to the method of Strathmann (1967).

All algae were cultured with f/2 medium (Guillard, 1975) at 60 μmol m$^{-2}$ s$^{-1}$ under a dark/light cycle of 12 h:12 h at a temperature of 15 ± 1 °C. Algae were fed to copepods during their exponential growth phase. Copepods *C. sinicus* were collected from Jiaozhou Bay, Qingdao, China (120°8′ E, 36°8′ N) in April and May 2011 using a 0.5 m standard ring net equipped with a 160 μm mesh and a solid cod end. The copepods were grown at 15 ± 1 °C in 30 PSU sterilized seawater. In the incubation experiment, adult copepods were utilized as experimental animals.

### 2.2.2 Ingestion configuration

Four species of algae (*C. curvisetus*, *I. galbana*, *Gymnodinium* sp., and *E. huxleyi*) were utilized to assess dietary effects on changes in ingestion rate (IR), clearance rate (CR), and DMS/DMSP production in a salinity of 30 PSU. *I. galbana* was used




as a single food source, which served as a good-quality food, and four concentrations of *I. galbana* (10, 15, 20, and $25 \times 10^4$ cells $mL^{-1}$) and four levels of salinities (20, 25, 30, and 35 PSU) were set as concentration and salinity contrasts to obtain ingestion and DMS/DMSP information from food concentration and salinity. Single-factor experiments were performed, and all parameters were run individually.

### 2.2.3 IR and CR

Batches of 10 individual copepods in each of 3 replicates were sorted out and transferred to 250 mL polycarbonate bottles. Three bottles containing no grazers were used as controls. All bottles were topped off with suspensions of algae, sealed with parafilm, and fastened onto a spinning plankton wheel (2 rpm) at $15 \pm 1$ °C in darkness for 24 h. After a 24-h incubation, copepods and fecal pellets were harvested according to the methods of Tang (2001). Water samples were used to analyze cell density, IR, CR, DMS, $DMSP_p$, and $DMSP_d$. The IRs and CRs were calculated according to the equations of Frost (1972). Duplicates of 10 mL aliquots of each algal suspension were placed in 40 mL serum bottles that contained 2 mL of 10 mol $L^{-1}$ KOH solution for DMSP detection. Ten copepods from each bottle were placed individually in serum bottles for $DMSP_z$ (DMSP in the copepod body) measurement before gut clearance. Fecal pellets were separated from detritus with a mouth micropipette, rinsed with filtered seawater, and concentrated onto a 47 mm Whatman GF/F filter by gravity filtration to determine $DMSP_f$ (DMSP in the fecal pellets). All samples were stored at −70 °C until DMSP measurement.

### 2.2.4 DMS and DMSP determinations

According to the methods of Andreae and Barnard (1983), DMS and DMSP concentrations were determined using the purge-and-trap technique by a gas chromatograph, which was equipped with a flame photometric detector (Shimadzu GC-14B). For DMS measurement, 10 mL seawater was directly introduced into a glass purge chamber. Gravity filtering of samples for $DMSP_d$ was obtained according to the method of Kiene and Slezak (2006) with minor modifications. The filtrate and unfiltered seawater for $DMSP_d$ and total DMSP ($DMSP_t$, $DMSP_d + DMSP_p$) measurements were transferred to a 40 mL serum bottle containing 2 mL of 10 mol $L^{-1}$ KOH solution and kept at 4 °C for at least 24 h to complete cleavage. $DMSP_d$ and $DMSP_p$ concentrations were determined by the total DMS subtracting the DMS in original seawater. The $DMSP_f$ and $DMSP_z$ concentrations were calculated by subtracting DMS and $DMSP_d$ in filtered water.

### 2.3 Statistical analysis

Data were expressed as mean ± standard deviation. Student's *t* test and one-way ANOVA were used to determine the differences between control and treatment samples. Pearson correlations were utilized to assess the relationships between DMS/DMSP and IRs.





## 3 Results

### 3.1 Field experiment

#### 3.1.1 Abiotic parameter

The temperature and salinity in Jiaozhou Bay are described in Fig. 2A. The salinity changed from 29.7 PSU to 32.3 PSU,
and the lowest and highest salinities occurred in August 2010 (summer) and April 2011 (spring), respectively. The changing
range of temperature was 1.72 °C to 25.3 °C, and the lowest and highest temperatures were observed in December 2010
(winter) and August 2010 (summer), respectively.

#### 3.1.2 Biotic parameters

Chl $a$ concentrations in surface water ranged from 0.808 µg L$^{-1}$ to 11.0 µg L$^{-1}$, with an average value of 3.09 µg L$^{-1}$. Chl $a$
concentration peaked on April 2011 (spring) (Fig. 2B). The bacterial abundance in surface water ranged from $5.7 \times 10^7$ cells
L$^{-1}$ to $1.79 \times 10^8$ cells L$^{-1}$ (average of $9.68 \times 10^7$ cells L$^{-1}$), with a peak value in September 2010 (autumn) (Fig. 2B).

A total of 74 species of zooplankton were identified and were categorized into 5 phyla, 6 classes, 17 orders, 44 families,
and 55 genera. Among these species, 31 are Arthropoda, 22 are Coelenterata, 17 are planktonic larvae, 2 are Chaetognatha, 1
are Protozoa, and 1 are Tunicata (Table 2). The species composition of zooplankton varied with months, and the number of
species ranged from 19 to 35. The lowest and highest numbers of species occurred in February 2011 (winter) and June 2010
(summer), respectively. In total, 22 copepods were identified, and *Acartia pacifica*, *Acartia bifilosa*, *Centropages
abdominalis*, *Eurytemora pacifica*, and *C. sinicus* were the predominant copepods (Table 2 and Fig. 3C).

Temporal variations in the mean zooplankton abundance are shown in Fig. 3A. *Noctiluca scintillans* appeared as the
dominant zooplankton, contributing 91% of zooplankton abundance on June 2010. Copepods dominated the zooplankton
community during winter and spring and accounted for 39% to 83% of the total zooplankton (Fig. 3A). The changing trend
of copepod abundance showed that the minimum (7.5 individuals m$^{-3}$) occurred on July 2010, subsequently increased, and
peaked (596 individuals m$^{-3}$) on April 2011 (Fig. 3B). The *C. sinicus* abundance ranged from 0.143 individuals m$^{-3}$ to 123
individuals m$^{-3}$, and *C. sinicus* is the dominant copepod from October 2010 to May 2011. The *C. sinicus* abundance showed
a similar trend to that of copepod, and *C. sinicus* exhibited the highest and lowest abundances in spring and summer,
respectively (Figs. 3B and 3C).

#### 3.1.3 DMS/DMSP

Changes in DMS/DMSP are presented in Fig. 4. The average concentrations of DMSP$_d$, DMSP$_p$, and DMS in surface water
were 3.96 (1.14–9.93), 10.6 (3.29–21.2), and 1.91 (0.41–3.19) nmol L$^{-1}$, respectively. Compared with DMSP$_d$ and DMS,
DMSP$_p$ has higher concentrations at the same periods. The highest concentrations of DMSP$_p$, DMSP$_d$, and DMS in Jiaozhou
Bay were observed on September, September, and June 2010, respectively. In comparison, the lowest contents of DMSP$_p$,
DMSP$_d$, and DMS were observed on November 2010, April 2011, and April 2011, respectively.





### 3.1.4 Relationships with biotic and abiotic parameters

In Jiaozhou Bay, many factors (biotic and abiotic parameters) could affect DMS and DMSP concentrations, and five factors (salinity, temperature, Chl $a$ concentrations, bacterial abundance, and copepod abundance) were selected to investigate their effects on DMS and DMSP in this study. Salinity was negatively correlated with temperature and $DMSP_p$ ($p < 0.05$). In

comparison, Chl $a$ concentrations were positively correlated with the copepod abundance ($p < 0.01$), and the copepod abundance was positively correlated with *C. sinicus* abundance ($p < 0.05$). Furthermore, positive correlations were observed among DMS, $DMSP_d$, and $DMSP_p$ ($p < 0.01$). In addition, no significant correlation was observed between DMS/DMSP and zooplankton, copepod, *C. sinicus* abundance or Chl $a$ concentration ($p > 0.05$) (Table 3).

### 3.2 Incubation experiment

### 3.2.1 Dietary effects on IR, CR, and DMS/DMSP

After *C. sinicus* was fed on four different diets, IRs changed from $0.13 \times 10^3$ cells copepod$^{-1}$ h$^{-1}$ to $5.55 \times 10^3$ cells copepod$^{-1}$ h$^{-1}$ and CRs ranged from 0.26 mL copepod$^{-1}$ h$^{-1}$ to 0.80 mL copepod$^{-1}$ h$^{-1}$ (Fig. 5A). Four species of algae affected IRs and CRs of *C. sinicus* differently, with *C. curvisetus* being the optimum diet for the grazing of *C. sinicus* (IR = $5.55 \times 10^3$ cells copepod$^{-1}$ h$^{-1}$; CR = 0.80 mL copepod$^{-1}$ h$^{-1}$), *I. galbana* being the second, *E. huxleyi* being the third, and

*Gymnodinium* sp. being the fourth.

The ingestion of *C. sinicus* of four species of algae would result in different DMS productions (Fig. 5B). With regard to *I. galbana* and *C. curvisetus*, *C. sinicus* grazing promoted DMS production in the treatments compared with the controls, e.g., DMS in *C. curvisetus* treatment by *C. sinicus* grazing was 1.7-fold of DMS in the controls. In comparison, DMS production has an opposite changing trend for *E. huxleyi* and *Gymnodinium* sp., in which the treatments showed lower DMS production

than that in the controls.

For four algae species, $DMSP_p$ concentrations in the controls ranged from 48.78 nmol L$^{-1}$ to 7,165 nmol L$^{-1}$ and those in the treatments ranged from 27.25 nmol L$^{-1}$ to 11,055 nmol L$^{-1}$. In the control and treatment groups, $DMSP_p$ concentrations of *E. huxleyi* and *Gymnodinium* sp. were two to three orders of magnitude higher than those of *I. galbana* and *C. curvisetus* (Fig. 5C). $DMSP_p$ concentrations in the treatments with copepod exhibited diverse changes for four algae species, $DMSP_p$

concentrations for *I. galbana* and *C. curvisetus* decreased, and $DMSP_p$ concentrations for *E. huxleyi* and *Gymnodinium* sp. increased. $DMSP_d$ in the treatments were higher than those in the controls ($p > 0.05$). $DMSP_d$ concentrations in the controls ranged from 10.60 nmol L$^{-1}$ to 6,595 nmol L$^{-1}$. In comparison, $DMSP_d$ concentrations ranged from 11.39 nmol L$^{-1}$ to 10,848 nmol L$^{-1}$ in the treatments (Fig. 5D). $DMSP_d$ in the treatments were higher than those in the controls ($p > 0.05$).

Significant differences for the $DMSP_z$ and $DMSP_f$ contents of the four diets were observed (Fig. 5E). $DMSP_z$ contents

decreased according the following order: *E. huxleyi* > *Gymnodinium* sp. > *I. galbana* > *C. curvisetus*. $DMSP_z$ contents of *C. sinicus* fed on *E. huxleyi* were 23.51-fold of those fed on *C. curvisetus* (Fig. 5E). When compared with $DMSP_z$ contents, $DMSP_f$ of four diets had different contents with the following order: *Gymnodinium* sp. > *E. huxleyi* > *I. galbana* > *C.*



*curvisetus*. $DMSP_f$ contents of *C. sinicus* fed on *Gymnodinium* sp. were 126.3-fold of those fed on *C. curvisetus* (Fig. 5E). For *C. curvisetus*, *I. galbana*, *Gymnodinium* sp., and *E. huxleyi*, $DMSP_z$ and $DMSP_f$ accounted for 0.58%, 3.2%, 0.57%, 0.14% and 0.16%, 1.04%, 0.70%, 0.43% of the total amounts of DMS/DMSP ($DMSP_z + DMSP_f + DMS + DMSP_{d,p}$), respectively.

### 3.2.2 Food concentration experiment

When *I. galbana* concentrations increased, IRs gradually increased, peaked at $15 \times 10^4$ cells $mL^{-1}$, and subsequently declined (ranged from $2.45 \times 10^3$ cells $copepod^{-1}$ $h^{-1}$ to $5.12 \times 10^3$ cells $copepod^{-1}$ $h^{-1}$). In comparison, increased *I. galbana* concentrations induced the decrease in CRs, which ranged from 0.007 mL $copepod^{-1}$ $h^{-1}$ to 0.054 mL $copepod^{-1}$ $h^{-1}$ (Fig. 6A). DMS and $DMSP_p$ increased with the increase in *I. galbana* concentrations (Figs. 6B and 6C). DMS in the treatments were higher than those in the controls ($p < 0.05$). Moreover, DMS in the treatments (20.19–49.84 nmol $L^{-1}$) increased by 1.8% to 11% compared with that in the controls (18.23–48.96 nmol $L^{-1}$). $DMSP_p$ contents in the treatments (292.44–945.76 nmol $L^{-1}$) were lower than those in the controls (385.94–1,319.83 nmol $L^{-1}$) because of grazing activity.

$DMSP_d$ increased initially, peaked at $15 \times 10^4$ cells $mL^{-1}$, and decreased with the increase in *I. galbana* concentrations (Fig. 6D). $DMSP_d$ contents in the treatments (23.54–235.94 nmol $L^{-1}$) were higher than those in the controls (22.65–207.86 nmol $L^{-1}$) ($p > 0.05$), and those in the treatments increased by 97.95% compared with that in the controls ($20 \times 10^4$ cells $mL^{-1}$). A positive correlation between IR and the increment of DMS + $DMSP_d$ was observed ($r = 0.623$, $p = 0.377$, $n = 4$). In addition, a positive correlation between algal concentration and the decrement of $DMSP_p$ was discovered ($r = 0.767$, $p = 0.233$, $n = 4$).

$DMSP_z$ and $DMSP_f$ increased with the increase in *I. galbana* concentrations, peaked at $15 \times 10^4$ cells $mL^{-1}$, and declined (Fig. 6E). A significantly positive relationship was observed between IR and $DMSP_f$ ($r = 0.99$, $p = 0.01$, $n = 4$), accompanied by irrelevance between IR and $DMSP_z$ ($r = 0.07$, $p = 0.93$, $n = 4$). $DMSP_z$ and $DMSP_f$ of four *I. galbana* concentrations ($10 \times 10^4$, $15 \times 10^4$, $20 \times 10^4$, and $25 \times 10^4$ cells $mL^{-1}$) accounted for 1.9%, 4.5%, 3.8%, 2.5% and 3.3%, 2.4%, 1.4%, 1.2% of the sum of DMS/DMSP ($DMSP_z + DMSP_f + DMS + DMSP_{d,p}$) of *I. galbana*, respectively. We observed significant positive correlations between DMS and $DMSP_p$ ($r = 0.893$, $p = 0.003$, $n = 8$), accompanied by significant negative correlations between DMS and $DMSP_d$ ($r = -0.847$, $p = 0.008$, $n = 8$). In addition, significant negative correlations between $DMSP_d$ and $DMSP_p$ were noted in this study ($r = -0.804$, $p = 0.016$, $n = 8$).

### 3.2.3 Effects of salinity on IRs and DMS/DMSP production

The salinity experiments showed that IRs and CRs increased with the increase in salinities, peaked at 30 PSU (IR = $3.18 \times 10^3$ cells $copepod^{-1}$ $h^{-1}$; CR = 0.23 mL $copepod^{-1}$ $h^{-1}$), and declined (Fig. 7A).

In this study, Fig. 7B shows that increased salinities restrained the production of DMS. Moreover, DMS contents in the treatment decreased from 84.54 nmol $L^{-1}$ to 26.54 nmol $L^{-1}$ when salinities increased, which was consistent with the DMS changes in the control (reduction from 41.11 nmol $L^{-1}$ to 25.90 nmol $L^{-1}$). The $DMSP_d$ changes indicated a trend similar to





those of DMS contents, that is, increasing salinities decreased $DMSP_d$ contents. Fig. 7D shows that $DMSP_d$ contents decreased from 481.41 nmol $L^{-1}$ to 72.35 nmol $L^{-1}$ in the treatment. In comparison, $DMSP_d$ contents in the control reduced from 423.99 nmol $L^{-1}$ to 66.80 nmol $L^{-1}$. DMS contents of the treatments were higher than those of the controls ($p > 0.05$) and increased by 106%, 51%, 4.7%, and 2.4% at the salinities of 20, 25, 30, and 35 PSU, respectively.

The increasing salinities facilitated the accumulation of $DMSP_p$ in the controls and treatments. $DMSP_p$ contents in the treatments increased from 92.20 nmol $L^{-1}$ to 371.49 nmol $L^{-1}$, and those in the controls rose from 121.57 nmol $L^{-1}$ to 532.16 nmol $L^{-1}$. $DMSP_p$ contents in the controls and treatments at 30 PSU were 2.9- and 1.7-fold of those at 25 PSU, respectively. $DMSP_p$ contents in the treatments were lower than those in the controls because of grazing activity ($p > 0.05$). Furthermore, data analysis showed a positive correlation between IR and the decrement of $DMSP_p$ ($r = 0.945$, $p = 0.055$, $n = 4$).

$DMSP_d$ contents in the treatments were higher than those in the controls ($p > 0.05$) (Fig. 7D). The relationship between IR and the increment of DMS + $DMSP_d$ proved to be positive ($r = 0.662$, $p = 0.338$, $n = 4$). In comparison, the correlation between the increments of DMS and $DMSP_d$ was negative ($r = -0.955$, $p = 0.045$, $n = 4$).

$DMSP_z$ ranged from 0.21 nmol $L^{-1}$ to 5.38 nmol $L^{-1}$, reached the maximum content at 35 PSU, and reached the minimum content at 20 PSU. When salinities increased, $DMSP_z$ initially increased, subsequently declined, and finally increased (Fig.
7E). Maximum $DMSP_z$ contents at 35 PSU were 25.6-fold, 4.2-fold, and 24.5-fold of those at 20, 25, and 30 PSU, respectively. $DMSP_f$ had a similar trend to $DMSP_z$ in the salinity experiments, and its changing range was 0.78 nmol $L^{-1}$ to 3.44 nmol $L^{-1}$. $DMSP_f$ reached the minimum at 20 PSU and the peak at 25 PSU. Compared with 20, 30, and 35 PSU, $DMSP_f$ at 25 PSU increased by 341%, 98%, and 2%, respectively. At 35 PSU, $DMSP_z$ and $DMSP_f$ attained the maximum sum (8.75 nmol $L^{-1}$), which was 8.84-fold of the minimum at 20 PSU. With regard to DMS/DMSP (DMS + $DMSP_{d,p}$ + $DMSP_z$ +
$DMSP_f$) of *I. galbana*, $DMSP_z$ of the four salinities (20, 25, 30, and 35 PSU) accounted for 0.035%, 0.2%, 0.05%, and 1.1%, and $DMSP_f$ of the four salinities accounted for 0.13%, 0.56%, 0.41%, and 0.70%, respectively. Pearson correlation analysis did not detect a significant correlation between IR and $DMSP_z$ ($r = 0.37$, $p = 0.63$, $n = 4$), which coincided with the relationship between IR and $DMSP_f$ ($r = 0.30$, $p = 0.70$, $n = 4$). In addition, $DMSP_p$ was significantly negatively correlated with DMS ($r = -0.721$, $p = 0.044$, $n = 8$) and $DMSP_d$ ($r = -0.918$, $p = 0.001$, $n = 8$). In comparison, $DMSP_p$ was observed to
be positively correlated with $DMSP_f$ ($r = 0.491$, $p = 0.509$, $n = 4$) and $DMSP_z$ ($r = 0.705$, $p = 0.295$, $n = 4$).

## 4 Discussion

### 4.1 Effects of *C. sinicus* on DMS/DMSP in the field study

In Jiaozhou Bay, *C. sinicus* was the predominant copepod in 74 species of zooplankton identified from years 2010 to 2011. In this study, no significant correlations were observed between zooplankton, copepod, and/or *C. sinicus* abundance and
DMS/DMSP concentrations in the field study, illustrating that Jiaozhou Bay was a complex ecosystem with different abundances and types of phytoplankton, zooplankton, and/or copepod in the natural environment. Many kinds of copepods inhabited Jiaozhou Bay, and their IRs depended on copepod species, e.g., IRs of *Harpacticus* sp. (Yu et al., 2015) were 10-





fold of those of *C. sinicus* in this study. DMS/DMSP production was affected by phytoplankton species, which was difficult to investigate in the field study. We evaluated the effects of several agents (i.e., food, diet concentration, and salinity) on DMS and DMSP productions in the laboratory study. Our incubation data showed that copepod grazing increased DMS production, which was consistent with previous investigations on the effects of copepod grazing on DMS production (Dacey

and Wakeham, 1986; Yu et al., 2015). Consistent with our field study results, no significant correlations between mesozooplankton abundance and the distribution of DMS or $DMSP_d$ were also observed in the Gulf of Maine and St. Lawrence in previous studies (Cantin et al., 1996; Matrai and Keller, 1993). Cantin et al. (1996) concluded that mesozooplankton grazing played a minor role in DMS and $DMSP_d$ productions in the Gulf of St. Lawrence. The variable conditions in the natural environment can explain the reason for the minor role of zooplankton on DMS/DMSP production

and the inconsistent results of field and incubation studies.

**4.2 Dietary effects on copepod grazing and DMS/DMSP production**

Different diets contain various DMSP contents, which affected DMS production induced by copepod grazing. DMSP-rich algae (*E. huxleyi* and *Gymnodinium* sp.) were not the preferable food for copepod *C. sinicus* in the incubation study because the released acrylic acid from DMSP in algae prevents them from copepod grazing. Based on the species of the diets, algae

produced the corresponding DMS and DMSP ($DMSP_d$ and $DMSP_p$) contents, which might be different. For example, when comparing *Gymnodinium* sp. and *E. huxleyi* with *C. curvisetus* and *I. galbana*, we determined that cellular $DMSP_p$ in *Gymnodinium* sp. and *E. huxleyi* were one to two orders of magnitude higher than those in *C. curvisetus* and *I. galbana* (see Table 1). In terms of *I. galbana*, our detection results on cellular $DMSP_p$ production were consistent with those reported by Niki et al. (2000), indicating that cellular $DMSP_p$ production in given algae was approximately invariable.

When *Gymnodinium* sp. and *E. huxleyi* (DMSP-rich phytoplankton) were grazed by *C. sinicus*, acrylic acid or DMSP released from algae protected them from being grazed and stimulated the increase in $DMSP_p$. In the natural environment, acrylic acid or DMSP depressed copepod appetites and forced them to ingest other DMSP-poor phytoplankton. Thus, DMSP-poor *C. curvisetus* became the favorite diet of copepods among four algae species, elucidating that the *C. sinicus* fed on *C. curvisetus*, which evidently promoted DMS production in this study. Further studies have revealed that $DMSP_d$ was a

feeding inhibitor, and $DMSP_p$, $DMSP_d$, DMS, and acrylic acid constituted a cellular antioxidant system involved in the scavenging of hydroxyl radicals (Strom et al., 2003; Sunda et al., 2002). Non-DMS-producing phytoplankton species in a mixture of prey were preferentially selected by grazers, and single DMS/DMSP-rich diet decreased the food intake of copepods, e.g., copepod *Harpacticus* sp. had inferior IRs and PPRs when fed on DMS/DMSP-rich alga *Prymnesium parvum* (Wolfe et al., 1997; Wolfe and Steinke, 1996; Yu et al., 2015). In this study, DMSP-rich *Gymnodinium* sp. and *E. huxleyi*

repressed copepod grazing, elucidating the antipredator ability of these algae, which in turn reduced DMS/DMSP production.

Our results showed that DMSP in copepod bodies and fecal pellets accounted for 0.035% to 4.5% and 0.13% to 3.3% of DMS and DMSP in this study, illustrating that the ingestion of *C. sinicus* transferred DMSP from phytoplankton to the



copepod bodies and fecal pellets. When compared with *Harpacticus* sp., lower $DMSP_f$ and $DMSP_z$ contributions in *C. sinicus* attributed to the lower IRs of *C. sinicus*.

Based on the *C. sinicus* abundance on May 2011, the maximum $DMSP_z$ (0.02–7 nmol $L^{-1}$) and $DMSP_f$ (0.1–12 nmol $L^{-1}$) of *C. sinicus* in the seawater of Jiaozhou Bay were achieved. The results of our study and that of Tang (2001) showed that

$DMSP_p$ in copepod bodies and fecal pellets was an essential part of DMSP flux in the ocean. In addition, another sink of DMSP in the ocean was achieved by microbial processes. The results of Tang et al. (2001) and Dong et al. (2013) indicated that copepods and their pellets harbored a dense population of DCB (DMSP-consuming bacteria), which played an important role in DMSP degradation.

$DMSP_z$ and $DMSP_f$ contents were strongly associated with *C. sinicus* grazing. Their changing trends were consistent with

that of IR, which depended on the critical concentration of phytoplankton. IRs of *C. sinicus* increased steadily with the increase in algal concentration below the critical concentration and decreased above the critical concentration. Critical concentration differed depending on the copepod and algal species, which were confirmed by this study and other investigations (Yu et al., 2015).

### 4.3 Effects of salinity on copepod grazing and DMS/DMSP

In this study, low salinity induced high DMS production in seawater, whereas high salinity increased $DMSP_p$ (intracellular DMSP) and decreased $DMSP_d$, which were consistent with the conclusion of Variamuthy et al. (1985). DMSP, an osmolyte possessing cytomembrane permeability (Kirst, 1996), was excluded from cells when salinity decreased, illustrating the aforementioned DMSP changes. *C. sinicus* grazing promoted DMS production in this study, which was consistent with the results obtained by many investigators (Belviso et al., 1990; Christaki et al., 1996; Daly and DiTullio, 1996; Leck et al.,

1990). The decrement of $DMSP_p$ changed into DMS and $DMSP_d$ by copepod grazing, which was consistent with the positive relationships between IR and the increment of DMS + $DMSP_d$ and/or the decrement of $DMSP_p$. The increment of DMS was significantly negatively correlated with the increment of $DMSP_d$, indicating that the cleavage of $DMSP_d$ was the source of DMS by DMSP lyase. Many reports on the location of DMSP lyase in the cell have been published. Wolfe and Steinke (1996) indicated that the DMSP lyase location of *E. huxleyi* CCMP 370 was in the membrane bound inside cells. Stefels and

Dijkhuizen (1996) reported that DMSP lyase of *Phaeocystis* was membrane-bound and located extracellularly. Cellular locations and functions of DMSP lyase might differ depending on algal species.

This study confirmed that DMS production was affected by primary producers, e.g., algae. DMS/DMSP in algae transferred to the food web by predation, and several researchers investigated the effects of zooplankton grazing on DMS production with copepods and krill, indicating that the breakage of algal cells through sloppy feeding may increase $DMSP_d$

production (Dacey and Wakeham, 1986; Kaamatsu et al., 2004). When $DMSP_p$ concentrations changed, $DMSP_f$ and $DMSP_z$ concentrations were altered correspondingly, which was verified by other results in which the DMSP defecation rate of copepod *Acartia tonsa* feeding on *Tetraselmis impellucida* (prasinophyte) was closely related to food concentration and $DMSP_z$ content (Tang, 2001). Salinity changes altered the osmotic pressure surrounding copepod and algae cells, which in



turn adjusted DMSP (intracellular $DMSP_p$, DMSP in tissues, and DMSP in gut content). Copepods contained more DMSP at high salinity, indicating the osmoregulatory function of DMSP (Tang et al., 1999).

## 5 Conclusions

In the present study, field and incubation experiments were performed to investigate the effects of *C. sinicus* grazing on DMS production in Jiaozhou Bay. Copepods (*C. sinicus*) indicated no apparent effect on DMS/DMSP production in the field experiment. Appropriate diets and salinities facilitated DMS/DMSP production, e.g., *C. sinicus* feeding on *I. galbana* and *C. curvisetus* exhibited increased DMS production at 30 PSU in the laboratory experiment. *C. sinicus* grazing promoted the productions of DMS and $DMSP_d$, and DMS was released mainly from $DMSP_d$ and low salinity increased DMS production. Copepods and fecal pellets supplied substantial $DMSP_p$ into the water column and were important to the biogeochemical cycling of DMS.

*Competing interests*. The authors declare that they have no conflict of interest.

*Acknowledgements*. This work was financially supported by the National Key Research and Development Program of China (Grant No. 2016YFA0601302), the National Natural Science Foundation for Creative Research Groups (Grant No. 41521064), the Major International Joint Research Project of NSFC (Grant No. 41320104008), the AoShan Talents Program supported by the Qingdao National Laboratory for Marine Science and Technology (Grant No. 2015ASTP), and the National Natural Science Foundation of China (Grant No. 41776085). This study is MCTL Contribution No. 156.

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





**Table 1.** Sizes and DMSP of phytoplankton cells.

| Algae | Dimension (μm) | Biovolume (μm$^3$) | Carbon (pg cell$^{-1}$) | DMSP (fmol cell$^{-1}$) |
|---|---|---|---|---|
| *C. curvisetus* | 7.8 × 4.6 | $9.04 \times 10^1$ | $1.15 \times 10^1$ | $1.0 \times 10^{-1}$ |
| *I. galbana* | 6.0 × 5.8 | $1.11 \times 10^2$ | $2.05 \times 10^1$ | $4.0 \times 10^0$ |
| *Gymnodinium* sp. | 14.0 × 12.0 | $1.13 \times 10^3$ | $1.53 \times 10^2$ | $3.6 \times 10^1$ |
| *E. huxleyi* | 2.7 × 3.2 | $1.25 \times 10^1$ | $2.94 \times 10^0$ | $3.0 \times 10^1$ |



**Table 2.** Species of zooplankton in Jiaozhou Bay.

| Phylum | Species | 2010.06 | 2010.07 | 2010.08 | 2010.09 | 2010.10 | 2010.11 | 2010.12 | 2011.01 | 2011.02 | 2011.04 | 2011.05 |
|---|---|---|---|---|---|---|---|---|---|---|---|---|
| Protozoa | *Noctiluca scientillans* Kofoid et Swezy | 1 | 1 | 1 | 1 | 1 | 1 | 1 | 1 | 1 | 1 | 1 |
| Coelenterata | *Ectopleura dumontieri* (Van Beneden) | | | | | 1 | | | 1 | 1 | | |
| | *Bougainvillia muscus* (Allman) | 1 | | | | | | | | | | |
| | *Sarsia japonica* (Nagao) | 1 | 1 | 1 | | | 1 | | 1 | | 1 | 1 |
| | *Rathkea octopunctata* (M.Sars) | | | | | | | | 1 | 1 | | |
| | *Turritopsis nutricula* (McCrady) | | | | | | | | | 1 | | 1 |
| | *Euphysora bigelowi* Maas | | | | 1 | | | | | | | |
| | *Euphysora knides* Huang | 1 | | | 1 | | | 1 | | | | |
| | *Zanclea costata* (Linne) | | | 1 | | | 1 | | | | | |
| | *Obelia* spp. | 1 | | 1 | 1 | 1 | 1 | | | | 1 | 1 |
| | *Clytia hemisphaerica* (Linnaeus) | 1 | 1 | | 1 | 1 | 1 | 1 | 1 | 1 | 1 | 1 |
| | *Eucheilota menoni* Kramp | | | | | 1 | 1 | 1 | | | | |
| | *Eirene cylonensis* Browne | | | | | 1 | 1 | | | | | 1 |
| | *Liriope petraphylla* (Chamisso et Eysenhardt) | | | | | | 1 | | | | | |
| | *Sugiura chengshanense* (Ling) | 1 | | | | | | | | | | |
| | *Lovenella* sp. | | | | 1 | 1 | | | | | 1 | |
| | *Malagazzia carolinae* (Mayer) | 1 | 1 | | | | | | | | | |
| | *Proboscidactyla flavicirrata* Brandt | 1 | | | | | | | 1 | 1 | 1 | 1 |
| | *Aequorea conica* Browne | | | | | | | | 1 | | | |
| | *Eutima levuka* (Agassiz et Mayer) | | 1 | | 1 | | | | | | | |
| | *Muggiaea atlantica* Cunningham | | 1 | | 1 | | | | | | | |
| | *Pleurobrachia globosa* Moser | | | | 1 | 1 | | | | | | |
| | *Beröe cucumis* Fabricius | | | | | | 1 | | | | | |
| Arthropoda | *Penilia avirostris* Dana | | 1 | | | | | | | | | |
| | *Evadne nordmanni* Loven | | 1 | | | | | | | | | |
| | *Calanus sinicus* Brodsky | 1 | 1 | 1 | 1 | 1 | 1 | 1 | 1 | 1 | 1 | 1 |
| | *Paracalanus parvus* (Claus) | 1 | | | 1 | 1 | 1 | | 1 | | | 1 |
| | *Eurytemora pacific* Sato | | | | | | | 1 | 1 | | 1 | 1 |
| | *Pseudodiaptomus poplesia* Shen | | | | | | | 1 | 1 | | | |
| | *Sinocalanus tenellus* (Kikuchi) | | | | | | | 1 | 1 | | | |
| | *Centropages abdominalis* Sato | 1 | 1 | | | | 1 | 1 | 1 | 1 | 1 | 1 |
| | *Centropages dorsispinatus* Thompson et Scott | | | 1 | 1 | 1 | 1 | | | | | |
| | *Centropages tenuiremis* Thompson et Scott | 1 | 1 | 1 | 1 | 1 | | | | | | 1 |
| | *Calanopia thompsoni* A.Scott | 1 | 1 | 1 | 1 | 1 | 1 | | | | | |
| | *Labidocera pavo* (Dana) | | | 1 | | | | | | | | |
| | *Labidocera euchaeta* Giesbrecht | | | | 1 | | 1 | 1 | 1 | 1 | | |
| | *Labidocera bipinnata* Tanaka | 1 | 1 | 1 | 1 | 1 | 1 | | | | | 1 |
| | *Pontellopsis tenuicauda* (Giesbrecht) | 1 | | | | | | | | | | |
| | *Acartia bifilosa* Giesbrecht | 1 | | | 1 | 1 | 1 | | 1 | 1 | 1 | 1 |
| | *Acartia pacifica* Steuer | 1 | 1 | 1 | 1 | 1 | 1 | | | | | 1 |
| | *Tortanus dextrilobatus* Chen and Zhang | 1 | 1 | 1 | 1 | 1 | | | | | 1 | 1 |
| | *Tortanus spinicaudatus* Shen et Bai | | | 1 | 1 | | | | | | | |
| | *Tortanus forcipatus* (Giesbrecht) | | | | | 1 | 1 | | | | | |
| | *Oithona similis* Claus | | | | | 1 | | | 1 | | 1 | 1 |
| | *Corycaeus affinis* Mcmurrichi | | 1 | 1 | 1 | | 1 | | 1 | 1 | | |
| | Harpacticoida | | | | | | | 1 | 1 | 1 | 1 | |
| | *Monstrilla* sp. | | | | 1 | | 1 | | | | | |
| | *Themisto gracilipes* (Norman) | | | | | | | | | | 1 | 1 |
| | *Acanthomysis longirostris* Ii | | | | | 1 | 1 | 1 | 1 | 1 | 1 | 1 |
| | *Acetes japonicus* Kishinouye | 1 | | | | | | | | | | |
| | Gammaridea | 1 | | 1 | 1 | | 1 | 1 | | | 1 | 1 |
| | *Caprella* sp. | | | 1 | 1 | | | | | | | 1 |
| | *Microniscus* sp. | | | | 1 | 1 | | | | | | |
| | *Leueon* sp. | | | | | | | 1 | 1 | 1 | | |
| Chaetognatha | *Sagitta nagae* Alvarino | | | 1 | | 1 | 1 | 1 | | | | 1 |
| | *Sagitta crassa* Tokioka | 1 | 1 | 1 | 1 | 1 | 1 | 1 | 1 | 1 | 1 | 1 |
| Tunicata | *Oikopleura dioica* Fol | | | | | 1 | 1 | | | | | 1 |
| Planktonic larvae | Trochophore larva | 1 | | | | | | | | | | |
| | Polychaeta larva | 1 | | 1 | 1 | 1 | 1 | 1 | 1 | | | 1 |
| | Gastropod post larva | 1 | 1 | 1 | 1 | 1 | | | | | 1 | 1 |
| | Bivalve larva | 1 | | 1 | | 1 | 1 | | 1 | 1 | | 1 |
| | Nauplius larva (Copepoda) | | | | | 1 | | | | | 1 | |
| | Nauplius larva (Cirripedia) | 1 | | | | | | | | | 1 | |
| | Cypris larva | 1 | | | | | | | | | | |
| | Macrura larva | 1 | 1 | 1 | 1 | 1 | | | | | 1 | 1 |
| | Brachyura zoea larva | 1 | 1 | 1 | 1 | 1 | 1 | | | | 1 | 1 |
| | Megalopa larva | 1 | 1 | 1 | 1 | | | | | | | |
| | Porcellana zoea larva | 1 | 1 | 1 | 1 | | | | | | | |
| | Alima larva | 1 | 1 | 1 | | | | | | | | |
| | Ophiopluteus larva | | | | 1 | | | | | | | |
| | Echinopluteus larva | | 1 | | 1 | 1 | 1 | 1 | 1 | 1 | | |
| | Echinodermata larva | | | | 1 | | 1 | | | | | |
| | Fish eggs | 1 | 1 | | | | | | | | 1 | 1 |
| | Fish larva | 1 | 1 | 1 | | | | | 1 | 1 | 1 | 1 |

"1" means appearance.



**Table 3.** Pearson correlation analyses between DMS/DMSP and abiotic and biotic parameters.

| | | Temperature | Salinity | Chl a | Bacteria | Zooplankton | Copepod | C. sinicus | DMS | DMSP$_d$ | DMSP$_p$ |
|---|---|---|---|---|---|---|---|---|---|---|---|
| Temperature | r | 1 | | | | | | | | | |
| | p value | 0 | | | | | | | | | |
| Salinity | r | −0.678$^*$ | 1 | | | | | | | | |
| | p value | 0.022 | 0 | | | | | | | | |
| Chl a | r | −0.082 | 0.423 | 1 | | | | | | | |
| | p value | 0.811 | 0.195 | 0 | | | | | | | |
| Bacteria | r | 0.437 | −0.577 | −0.202 | 1 | | | | | | |
| | p value | 0.179 | 0.063 | 0.552 | 0 | | | | | | |
| Zooplankton | r | 0.243 | 0.379 | 0.441 | −0.116 | 1 | | | | | |
| | p value | 0.472 | 0.25 | 0.174 | 0.733 | 0 | | | | | |
| Copepod | r | −0.166 | 0.448 | 0.877$^{**}$ | −0.114 | 0.125 | 1 | | | | |
| | p value | 0.626 | 0.167 | 0 | 0.738 | 0.714 | 0 | | | | |
| C. sinicus | r | −0.225 | 0.382 | 0.315 | −0.387 | −0.027 | 0.698$^*$ | 1 | | | |
| | p value | 0.507 | 0.247 | 0.345 | 0.239 | 0.938 | 0.017 | 0 | | | |
| DMS | r | 0.302 | −0.353 | −0.264 | 0.104 | 0.385 | −0.47 | −0.329 | 1 | | |
| | p value | 0.367 | 0.287 | 0.434 | 0.762 | 0.242 | 0.144 | 0.323 | 0 | | |
| DMSP$_d$ | r | 0.454 | −0.588 | −0.227 | 0.56 | 0.117 | −0.241 | −0.219 | 0.812$^{**}$ | 1 | |
| | p value | 0.161 | 0.057 | 0.503 | 0.073 | 0.732 | 0.475 | 0.518 | 0.002 | 0 | |
| DMSP$_p$ | r | 0.477 | −0.603$^*$ | −0.103 | 0.538 | 0.218 | −0.205 | −0.229 | 0.787$^{**}$ | 0.900$^{**}$ | 1 |
| | p value | 0.138 | 0.049 | 0.762 | 0.088 | 0.521 | 0.546 | 0.499 | 0.004 | 0 | 0 |

$^*$, Correlation is significant at the 0.05 level (2-tailed).
5  $^{**}$, Correlation is significant at the 0.01 level (2-tailed).





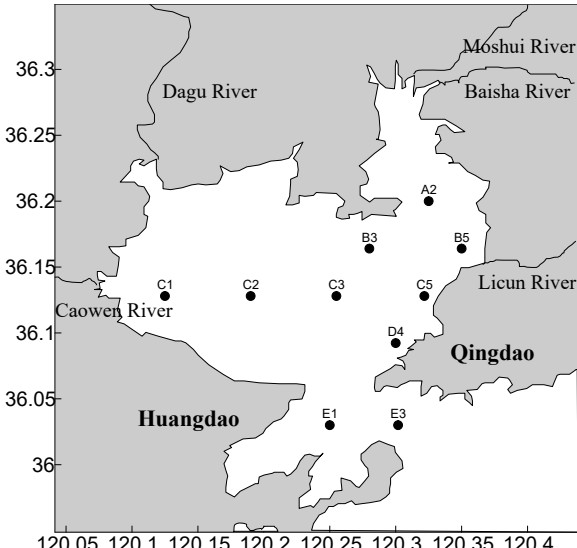

**Figure 1.** Location of fieldwork sampling stations in Jiaozhou Bay.





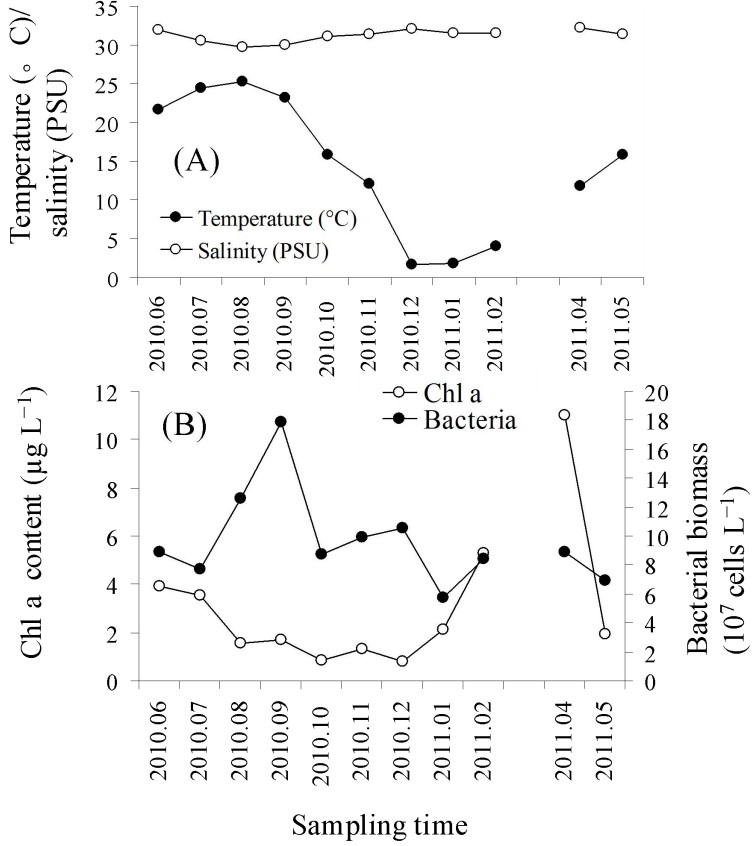

**Figure 2.** Monthly changes in temperature and salinity (A) and Chl *a* content and bacterial abundance (B) in Jiaozhou Bay.





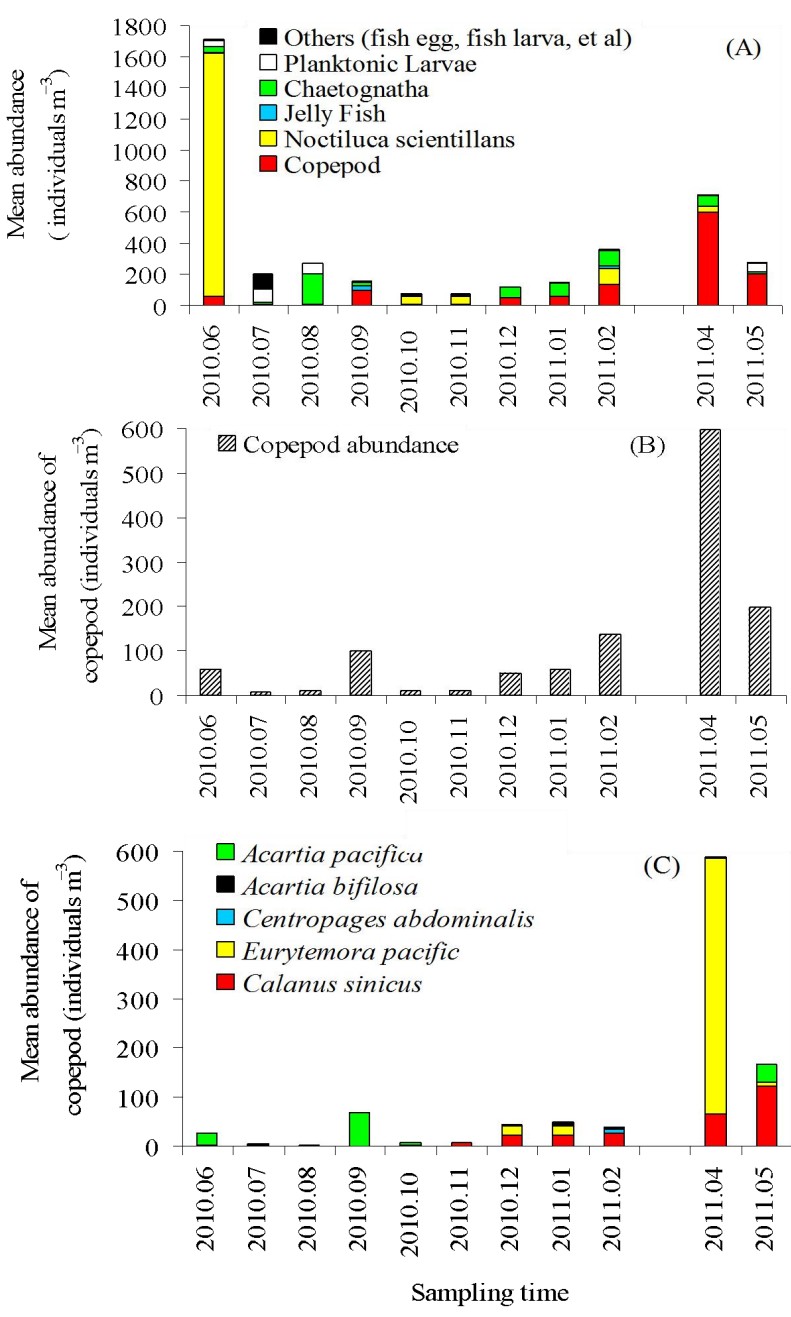

**Figure 3.** Monthly changes in the mean abundances of zooplankton (A), copepod (B), and predominant copepod (C) in Jiaozhou Bay.





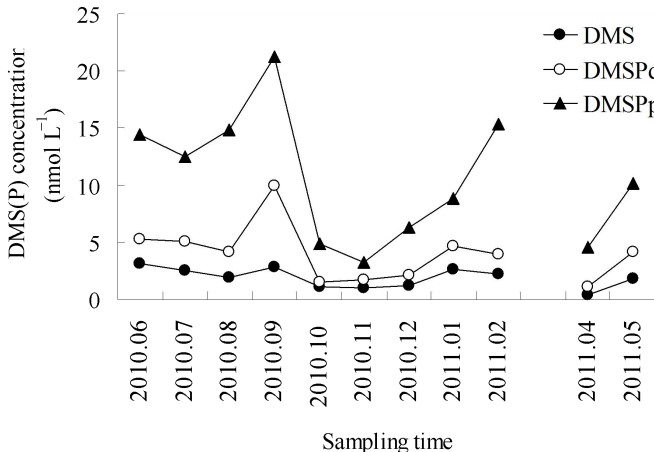

**Figure 4.** Mean DMS and DMSP concentrations in surface seawater of 10 stations in Jiaozhou Bay.



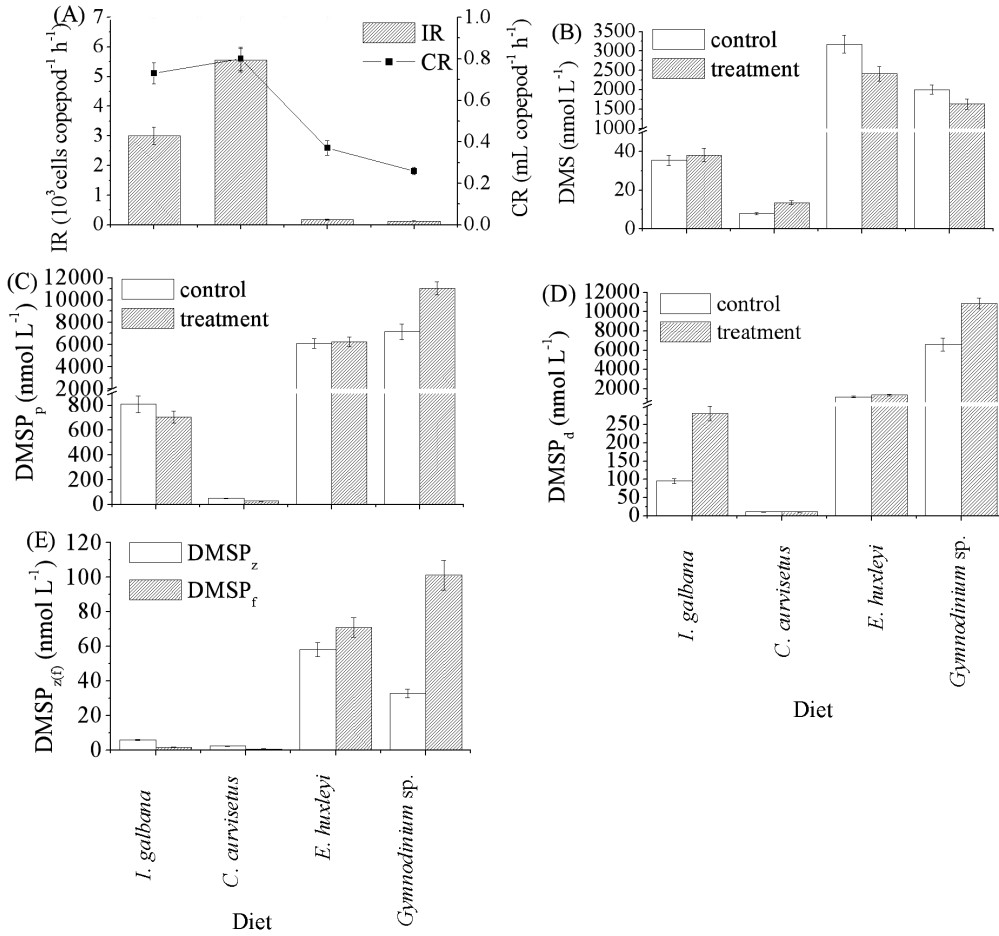

**Figure 5.** Effects of *Calanus sinicus* grazing on IR and CR (A), DMS (B), DMSP$_p$ (C), DMSP$_d$ (D), and DMSP$_{z,f}$ (E) when they preyed on different diets. Error bars represent the standard deviation ($n$ = 3).



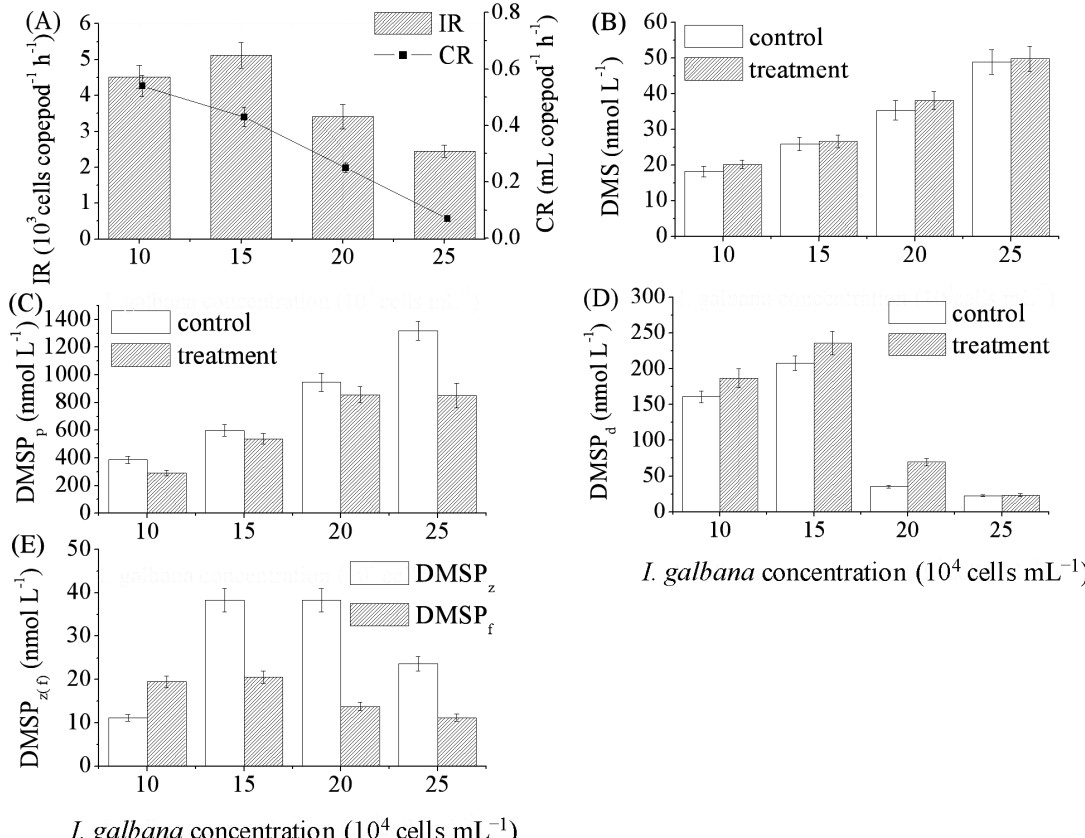

**Figure 6.** Effects of *C. sinicus* grazing on IR and CR (A), DMS (B), $DMSP_p$ (C), $DMSP_d$ (D), and $DMSP_{z,f}$ (E) when they preyed on different concentrations of *I. galbana*. Error bars represent the standard deviation ($n = 3$).



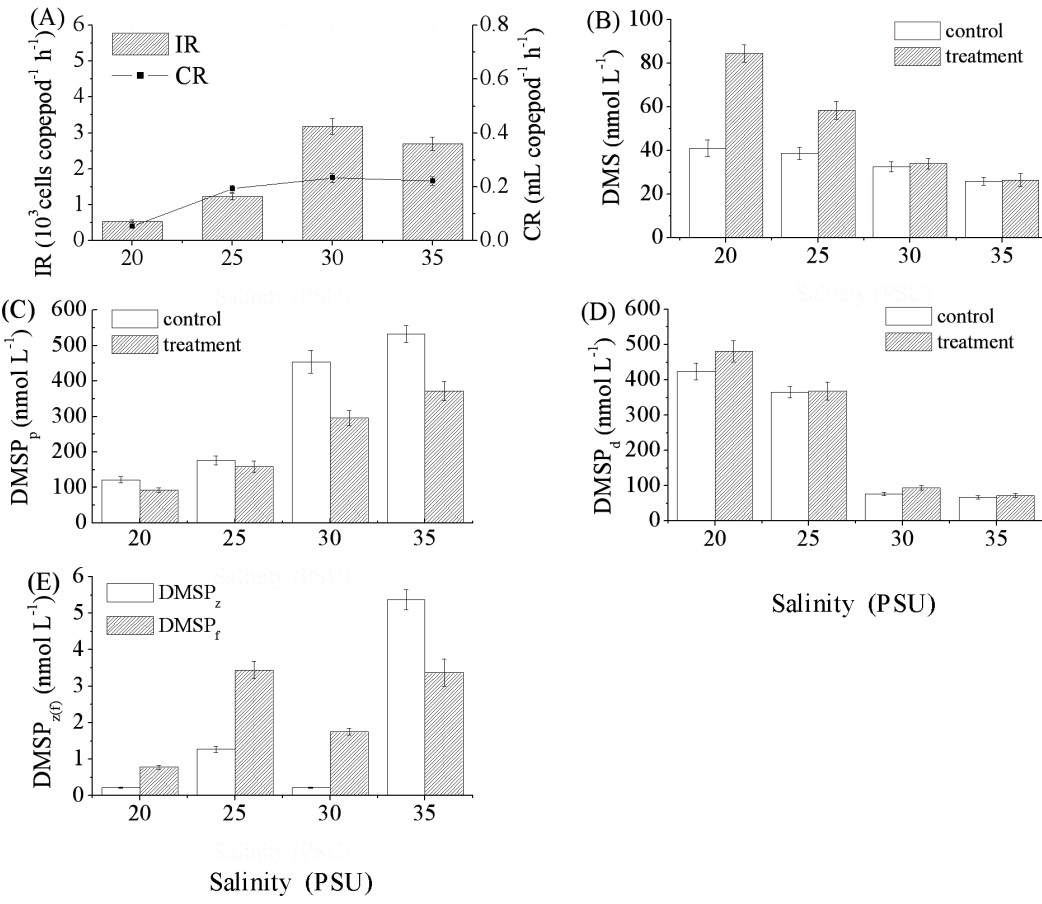

**Figure 7.** Effects of *C. sinicus* grazing on IR and CR (A), DMS (B), DMSP$_p$ (C), DMSP$_d$ (D), and DMSP$_{z,f}$ (E) when they preyed at different salinities. Error bars represent the standard deviation ($n = 3$).

