# Peer review of "Role of *Calanus sinicus* (Copepoda, Calanoida) on dimethylsulfide production in Jiaozhou Bay"

_Biogeosciences, 2017_

## Referee Comment (RC1) · Anonymous Referee #1 · 16 Feb 2018

This work looked at the relationship between copepod grazing and DMS(P) production. Laboratory experiments were conducted using the copepod Calanus sinicus and four phytoplankton species with varying morphologies and intracellular DMS(P) concentrations. Field measurements of zooplankton species and abundances, and DMS(P) concentrations were also conducted at monthly intervals. For reasons outlined below, I recommend that the authors separate the lab and field measurements and focus on publishing the lab studies in a journal such as Marine Ecology Progress Series.

Major comments (1) The major issue associated with this manuscript is the design of the field study. It is not clear to me why the authors would measure DMS(P) and zooplankton species composition and abundance, in order to determine the influence of grazing. I would have thought that dilution grazing experiments (see the work of Mike

Landry) are an appropriate method to look at the effect of grazing on DMS(P). The identification and abundance of zooplankton are insufficient to determine their relevance to water-column DMS(P) concentrations. The alternative would have been to measure phytoplankton composition/abundance. (2) What is the motivation for varying salinity in some lab experiments? How will this affect the intracellular DMS(P) concentrations of the phytoplankton if it is an osmolyte? Were the DMS(P) concentrations measured at the different salinities? (3) In my experience, copepods will pretty much eat anything if they are hungry enough. Of course, this will have a big effect on IR and CR. Did you starve the grazers prior to adding the prey phytoplankton? (4) If you ever repeat the laboratory grazing experiments, you could include a treatment with antibiotics? This will inhibit any bacteria that metabolize DMS and you could see how relevant they are.

Smaller comments Page 1, Line 13 The field work should be referred to as measurements and not experiments. Page 1 Line 27 Remove this 'recently came under close scrutiny'. Page 3 Line 8 remove 'a conductivity–temperature–depth probe' Page 3 Line 9 Waterman or Whatman? Page 4 Line 1 what is meant by 'which served as a good-quality food,' Page 4 Line 10 I don't know the equations of Frost (1972) so some description is needed. To measure IR you presumably measure algal abundance before and after grazing? Page 4 Line 15 why were samples stored at -70oC? I wasn't aware that this is part of the typical DMS(P) protocol Page 5 Line 9 Report chlorophyll concentrations to 1 decimal place Page 5 Line 28 I suspect DMS(P) concentrations should also be reported to 1 decimal place Page 5 Line 30 Replace contents with concentrations Page 5 Line 16 Change 'would result' to 'resulted in'

Table 3 Why show correlations which are not significant?

[Figure]

---

## Referee Comment (RC2) · Anonymous Referee #2 · 2 May 2018

The manuscript “Role of Calanus sinicus (Copepoda, Calanoida) on dimethylsulfide production in Jiazhou Bay” by Juan et al attempts to understand the role of copepod Calanus sinicus in DMS production in Jiazhou Bay through insitu observations and lab experiments. The authors followed a yearly cycle of insitu observations on temperature, salinity, Chl a, TBC, zooplankton enumeration and speciation, DMS, DMSPp and DM-SPd at 10 stations in the Jiazhou Bay. They also performed lab experiments wherein they conducted zooplankton grazing experiments on select phytoplankton species to see the impact on DMS production. Though the hard work put in by the authors is commendable, there is a major disconnect between observations and lab experiments. Their observation on DMSP transfer from phytoplankton to copepod body, fecal pellet to seawater is not new and has been proposed quite some time back (Tang et al;

Belviso et al). The authors mention that data from the field measurements showed that Calanus sinicus did not have any apparent effect on DMS/DMSP production and then the authors go ahead to perform a complex grazing experiment to see the impact of grazing on DMS production. If the field data did not show any connect, what was the aim to perform the lab experiments? Perhaps the authors should have checked the gut content to see what species the copepod preferred to feed on? This might have given some clues on how to proceed. Also, the insitu observation part is not clear. Some of the concerns with regards to this work is jotted below. The authors mention time series sampling at 10 locations, but figure 2 shows data for only one site, which is this site? Or is this averaged data? If averaged then include standard deviation. What is the reason for the increase in DMSPp&d (and marginal increase in DMS) during September 2010? What is the major phytoplankton species during spring? As this might have answered the high DMSP observed during that time. In terms of copepod, the authors mention Calanus sinicus as the predominant copepod, but that does not seem to be the case as Eurytemora pacific was also dominant during three sampling with April 2011 showing maximum abundance. In the feed (diet) experiment it is clear that the copepods prefer I. galbana and C. curvisetus compared to E. Huxleyi and Gymnodinium sp. There is not much difference in DMS production in the treatment when compared to the control. On the contrary DMS production dropped in the treatment in the case of E. Huxleyi and Gymnodinium sp. in comparison to I. galbana and C. curvisetus which showed marginal increase in DMS production. More queries and corrections are mentioned below. Line 26: . . .in May 2011) and had no apparent . . . .. Introduction: Line 37-38: The authors mention that there was close scrutiny on DMSP, what kind of scrutiny, please elaborate. Line 48: what kind of biotic and abiotic factors? Elaborate. Line 49: replace 'account' with 'consume' Line 51: the authors mentioned that role of zooplankton grazing on DMSP biogeochemical processes are scarce. And later in the same paragraph include detailed studies on the impact of zooplankton grazing on DMS production. The aim for carrying out the grazing work needs to clearly spelt out. Line 62: delete 'Brodsky' Line 65: what field experiments were performed? Are the

authors referring to field measurements? Materials and methods: The authors mention collection of samples from 10 stations. I assume these are surface samples? How were the samples collected? Niskin sampler or any other sampler? Zooplankton samples were collected by vertical tows, mention depth or range of depth from where the vertical tows were done Line 91: spelling correction 'Whatman' GF/F filters There are many grammatical errors in the manuscript. Only a few are pointed out. The authors need to correct these. Line 334: Did the authors measure acrylic acid as a deterrent against grazing? Line 345-346: 'In this study,.......which in turn reduced DMS/DMSP production'. This does not seem to be the case as DMS production was high in Gymnodinium sp and E. Huxleyi as seen from control and did not depend on grazing. Figure 1: Include latitude and longitude or specify 'N' and 'E' Figure 3: (B) may be deleted as species wise in shown in (C) One of the important parameters that this work lacks is phytoplankton speciation of the natural samples. In the absence of that data, understanding DMSP variation becomes difficult. Grazing by zooplankton on phytoplankton is an important part which results in DMSP going from particulate (within cell) to dissolved (outside) and further the action by DMSP lyase (both by phytoplankton as well bacterial lysis) results in high DMS production. Grazing is studied by looking at the gut content or by isotopic work, neither being done in the present study, it's difficult to address DMS production to grazing. And finally, there is a complete disconnect between field results and the basis for performing grazing experiments in the laboratory. Thus, I do not recommend the manuscript for publication in its present form.

---

## Author Comment (AC1) · 12 Jun 2018

**Author's response for Referee #1:**

This work looked at the relationship between copepod grazing and DMS(P) production. Laboratory experiments were conducted using the copepod *Calanus sinicus* and four phytoplankton species with varying morphologies and intracellular DMS(P) concentrations. Field measurements of zooplankton species and abundances, and DMS(P) concentrations were also conducted at monthly intervals. For reasons outlined below, I recommend that the authors separate the lab and field measurements and focus on publishing the lab studies in a journal such as Marine Ecology Progress Series.

Response: In order to find the effects of copepod *Calanus sinicus* on DMS production in Jiaozhou Bay, we investigate the DMS(P) concentration and the zooplankton including copepod *Calanus sinicus* abundances in situ, according to the comments of the referee, dilution experiment were added in the revised manuscript, which has been done in the experiment from June 2010 to May 2011 to evaluate the role of zooplankton on DMS(P) production, and the laboratory experiment further investigated the effects of the impact factors (such as different food, salinity and food concentrations) on the copepod grazing and therefore DMS production.

Major comments

(1) The major issue associated with this manuscript is the design of the field study. It is not clear to me why the authors would measure DMS(P) and zooplankton species composition and abundance, in order to determine the influence of grazing. I would have thought that dilution grazing experiments (see the work of Mike Landry) are an appropriate method to look at the effect of grazing on DMS(P). The identification and abundance of zooplankton are insufficient to determine their relevance to water-column DMS(P) concentrations. The alternative would have been to measure phytoplankton composition/abundance.

Response: According to the previous study (Yu et al., 2015), we found that the DMS(P) release varied

depending on the species and the abundance of copepod, so we measured DMS(P) concentrations and zooplankton species composition and abundance to declare the relationships between DMS(P) concentrations and zooplankton species composition or abundance. From the data, we can find that copepod *Calanus sinicus* is the dominant copepod in Jiaozhou Bay. On the other hand, we know that zooplankton grazing can release the DMSP in algae cell. The referee's comment is right, the identification and abundance of zooplanktonare are insufficient to determine their relevance to water-column DMS(P) concentrations. In fact, the dilution grazing experiments had been done in situ to evaluate the effect of zooplankton grazing on DMS(P) from June 2010 to May 2011, we did not put the data in the previous submission because that copepod numbers were usually low ($< 1$ $L^{-1}$) compared with microzooplankton. According to the referee's comments, we have put these data into the revised manuscript (See 2.1.3, 3.1.3, Table 3, Fig. 4.). We did not measure phytoplankton composition/abundance, while Zheng et al (2014) and Luo et al (2016) investigated the species composition and abundance of phytoplankton in 2010 and 2011 in Jiaozhou Bay, respectively. Therefore, we cited their data in order to understand DMSP variation in Jiaozhou Bay (see 4.1, Table 4, and Fig. 9) .

(2) ①What is the motivation for varying salinity in some lab experiments? ②How will this affect the intracellular DMS(P) concentrations of the phytoplankton if it is an osmolyte? ③Were the DMS(P) concentrations measured at the different salinities?

Response: ① Firstly, we found that the salinity in Jiaozhou Bay varied depending on different months; Secondly, multiple laboratory studies have confirmed that salinity can significantly influence the ingestion rates of copepod and the S-compounds production of phytoplankton (Tang et al., 1999; Yu et al., 2015). Therefore, varying salinities were set up in lab experiments to investigate the effects of

salinity on copepod grazing and the variations of DMSP and DMS concentrations. See 2.2.2.

② In our study, DMS and $DMSP_d$ (extracellular DMSP) concentrations decreased with salinity increase; conversely, the higher salinity stimulated the $DMSP_p$ (intracellular DMSP) accumulation (Fig. 8), which was in accordance with observation of a benthic diatom documented by van Bergeijk et al. (2003) and *Skeletonema costatum* documented by Yang et al. (2011). When the salinity went up or down, intracellular DMSP, as an osmotically active compound, was accumulated or released in order to help algal cells adjust their osmotic potential (Kirst, 1996). See 4.3.

③ Yes. The algae *Isochrysis galbana* and the copepod *Calanus sinicus* were cultured in different salinities, and the DMS(P) samples were get from the varying salinity environment and then they were measured, see Fig. 8.

(3) In my experience, copepods will pretty much eat anything if they are hungry enough. Of course, this will have a big effect on IR and CR. Did you starve the grazers prior to adding the prey phytoplankton?

Response: Yes, we starve the grazers prior to adding the prey phytoplankton, and the sentence "In order to ensure that the copepods were hungry to graze the diets, copepods were starved for 24 h before they were fed with phytoplankton." was added in the manuscript, see Page 5 Lines 2-3. In spite of starvation, copepods can detect and react to plumes of DMS (Steinke et al., 2006), and the released acrylic acid from DMSP in algae prevents them from copepod grazing. Thus, although they are hungry enough they also can detect and react to the DMSP-rich or DMSP-poor algae, and then they will eat more for DMSP-poor algae and less for DMSP- rich algae.

(4) If you ever repeat the laboratory grazing experiments, you could include a treatment with antibiotics? This will inhibit any bacteria that metabolize DMS and you could see how relevant they

are.

Response: According to referee's comments and the methods of Agostini et al. (2016), we have repeated the laboratory grazing experiments and the antibiotics (0.025 g $L^{-1}$ penicillin G potassium + 0.08 g $L^{-1}$ streptomycin sulphate + 0.04 g $L^{-1}$ neomycin sulphate) were used to inhibit the bacteria in the copepod culture, and the relevant were analyzed in the paper. The following was added in the revised manuscript:

1) 'According to the methods of Wolfe and Steinke (1996), the algal culture was detected by epifluorescence microscopy following staining with acridine orange and by plating on 1% peptone agar plates to check for bacterial growth. No bacterial contamination was found in any of the experimental cultures.' See Page 4 Lines 14-16. 2) 'Copepods were rinsed with sterilized seawater before the beginning of the grazing experiment.' See Page 5 Line 3. 3) 'We ran preliminary experiment to check the effects of the bacteria on DMS concentration. The treatment with antibiotics (0.025 g $L^{-1}$ penicillin G potassium + 0.08 g $L^{-1}$ streptomycin sulphate + 0.04 g $L^{-1}$ neomycin sulphate) were used to inhibit the bacteria in the algal culture. When *C. sinicus* were fed on the four diets (*I. galbana*, *C. curvisetus*, *E. huxleyi*, *Gymnodinium* sp.), no significant differences were found between DMS concentrations in the control (without antibiotics) and those in the treatment (with antibiotics) (data no shown). Therefore, the copepod cultures were not treated with antibiotics in our laboratory experiment to obtain axenicity. Yost and Mitchelmore (2009) reported that antibiotic treatment negatively affected algal growth, which was the other reason for not using antibiotics." See Page 5 Lines 22-29.

Smaller comments:

(5) Page 1, Line 13 The field work should be referred to as measurements and not experiments.

Response: The sentence has been replaced with "in field measurements and laboratory experiments",

see Page 1 Line 13.

(6) Page 1 Line 27 Remove this 'recently came under close scrutiny'.

Response: 'recently came under close scrutiny' has been removed, see Page 1 Line 27.

(7) Page 3 Line 8 remove 'a conductivity–temperature–depth probe'

Response: 'a conductivity–temperature–depth probe' has been removed, see Page 3 Line 29.

(8) Page 3 Line 9 Waterman or Whatman?

Response: 'Whatman' is correct, and 'Waterman' has been replaced with 'Whatman', see Page 3 Line 30.

(9) Page 4 Line 1 what is meant by 'which served as a good-quality food,'

Response: 'which served as a good-quality food,' means 'which was a DMSP-poor and favorite food for *C. sinicus*,', and the sentence has been revised, see Page 4 Line 24.

(10) Page 4 Line 10 I don't know the equations of Frost (1972) so some description is needed. To measure IR you presumably measure algal abundance before and after grazing?

Response: Yes, the initial and final algal concentrations should be measured to abtain IR and CR. The description of the equations of Frost (1972) has been added in the revised manuscript as follows. 'The IRs and CRs were calculated according to the equations of Frost (1972). Because no significant differences were found in algae concentrations between the initial and final control bottles, the growth constant ($k$) for algal growth was eliminated from the equations, thus yielding:

$$CR = \frac{V \times \ln(C_1 / C_2^*)}{Nt}$$

(1)

$$IR = CR \times C_1$$

(2)

Where $CR$ is the clearance rate (mL $ind^{-1}$ $h^{-1}$), $IR$ is the ingestion rate (cells $ind^{-1}$ $h^{-1}$), $C_1$ is the initial algal concentration in control bottles (cells $mL^{-1}$), $C_2^*$ is the final algal concentration in the

experimental bottles (cells mL$^{-1}$), $t$ is the duration of the experiment (h), $V$ and $N$ are the volume (mL) and number of copepods in the experimental bottles (ind), respectively.' See 2.2.3.

(11) Page 4 Line 15 why were samples stored at -70°C? I wasn't aware that this is part of the typical DMS(P) protocol

Response: The referee's comment is right, and this is not part of the typical DMS(P) protocol. The sentence 'All samples were stored at −70 °C until DMSP measurement.' has been deleted, see Page 5 Line 21.

(12) Page 5 Line 9 Report chlorophyll concentrations to 1 decimal place Page 5 Line 28 I suspect DMS(P) concentrations should also be reported to 1 decimal place

Response: chlorophyll concentrations and DMS(P) concentrations were all changed into 1 decimal place , see Page 6 Line 22 and Page 7 Line 23.

(13) Page 5 Line 30 Replace contents with concentrations

Response: 'contents' has been replaced with 'concentrations', see Page 7 Line 25.

(14) Page 6 Line 16 Change 'would result' to 'resulted in'

Response: 'would result' has been changed to 'resulted in', see Page 8 Line 11.

(15) Table 3 Why show correlations which are not significant?

Response: According to the comments of the referee, Table 3 has been deleted, and the correlations which are significant were described in the text.

---

## Author Comment (AC2) · 12 Jun 2018

[revised manuscript text omitted]

**2.1.2 Abundance and taxonomic composition of zooplankton**

Zooplankton samples were collected by vertical tows from the bottom to the surface (the depth varied from 3 m to 28 m according to different stations) using a conical–cylindrical plankton net with a 50 cm mouth opening and a 160  $\mu$ m mesh size. A flow meter was used to estimate the amount of filtered water. We retrieved the net at 0.3 m s-1 to 0.5 m s-1 at each station.

5 Samples were then rinsed into cod-end buckets, concentrated, and preserved in 5% formalin for qualitative and quantitative analyses of in situ zooplankton. Taxonomy was determined by optical microscopy. In situ abundance was calculated using tow volume estimates, which were determined by net dimensions and flow meter values.

**2.1.3 Dilution experiments**

Dilution experiments were set up according to the methods of Landry and Hassett (1982) and Wolfe et al. (2000), which are widely used to estimate microzooplankton grazing and phytoplankton growth rates. The technique assumes that increasingly diluted treatments reduce grazer-prey encounter and therefore grazing rates (g), without changing specific growth rates ( $\mu$ ) of prey. Net production of a prey biomaker B is thus given by  $B(t) = B(0)e^{(\mu-dg)t}$ , where d is the fraction of unfiltered water; regressing 1/t ln (B(t)/B(0) vs. d yields  $\mu$  as the Y-intercept and g as the negative of the slope (Wolfe et al., 2000).

- Depending upon the time allowed for water collection, we conducted dilution experiments aboard at three stations (C3: 120.26°E, 36.13°N; D4: 120.30°E, 36.09°N; E3: 120.30°E, 36.03°N) on each cruise from June 2010 to May 2011 in Jiaozhou Bay (Fig. 1). A dilution series was prepared, consisting of 100, 80, 60, 40, and 20% unfiltered water in 1 L polycarbonate bottles (washed with 10% HCl and distilled water, and rinsed with seawater prior to use). The water was collected using 12 L Niskin bottles on a CTD rosette. Water for dilutions was filtered using gravity through Gelman Suporcap 0.2 µm capsule filters into an acid-washed carboy. These bottles were incubated for 24 h under simulated in situ conditions in a water-bath deck incubator with neutral density screening. Nutrients were added in the bottles to ensure
- constant phytoplankton growth in the dilution series when the final concentrations of nitrate and phosphate were lower than 0.5  $\mu$ M and 0.03  $\mu$ M. We did not pre-screen water to remove mesozooplankton grazers, because copepod numbers were usually low (< 1 L-1) compared with microzooplankton and our preliminary trials showed that mesograzers had a negligible
- 25 grazing impact at natural densities compared to microzooplankton. Specific growth and grazing rates of Chl *a* were calculated by regressing the time-normalized, log-transformed ratio of final and initial concentrations vs. fraction unfiltered water (Landry and Hassett, 1982).

**2.1.4 Abiotic parameters and analyses of chlorophyll a (Chl a) and bacteria**

The shipboard measurements of temperature and salinity of surface water were obtained with the Sea-Bird 911. The surface water samples were collected by a Niskin sampler and then they were filtered using Whatman GF/F filters (nominal pore size of 0.7 μm) to determine Chl *a*. Chl *a* concentrations were measured using a Hitachi F4500 fluorometer according to the

methods of Parsons et al. (1984). Based on the procedures of Porter and Feig (1980), bacteria were counted by epifluorescence microscopy (Hitachi F4500; total magnification of  $\times 1,000$ ).

**2.2 Incubation experiment**

**2.2.1 Phytoplankton and zooplankton cultures**

- 5 Phytoplankton species (*Isochrysis galbana, Chaetoceros curvisetus, Emiliania huxleyi*, and *Gymnodinium* sp.) were provided by the Marine Microalgae Research Center, Ocean University of China in different sizes and DMSP contents. They are widespread species in Jiaozhou Bay and Chinese coastal waters (Li et al., 2005; Zhong et al., 2001). These algae are described in Table 1, and the sizes of algal cells are obtained by measuring at least 200 cells using a calibrated ocular micrometer with a light microscope at a magnification of × 400. Cell volume was estimated by approximating the shape of
- 10 each species to an elliptic sphere, which subsequently determined the biovolume according to the method of Verity et al. (1992). In this study, we determined the cellular carbon contents in phytoplankton according to the method of Strathmann (1967).

All algae were cultured with f/2 medium (Guillard, 1975) at 60  $\mu$ mol m-2 s-1 under a dark/light cycle of 12 h:12 h at a temperature of 15 ± 1 °C. According to the methods of Wolfe and Steinke (1996), the algal culture was detected by

- 15 epifluorescence microscopy following staining with acridine orange and by plating on 1% peptone agar plates to check for bacterial growth. No bacterial contamination was found in any of the experimental cultures. Algae were fed to copepods during their exponential growth phase. Copepods *C. sinicus* were collected from Jiaozhou Bay, Qingdao, China (120°8' E, 36°8' N) in April and May 2011 using a 0.5 m standard ring net equipped with a 160 µm mesh and a solid cod end. The copepods were grown at 15 ± 1 °C in 30 PSU sterilized seawater. In the incubation experiment, adult copepods were utilized
- 20 as experimental animals.

**2.2.2 Ingestion configuration**

Four species of algae (*C. curvisetus*, *I. galbana*, *Gymnodinium* sp., and *E. huxleyi*) were utilized to assess dietary effects on changes in ingestion rate (IR), clearance rate (CR), and DMS/DMSP production in a salinity of 30 PSU. *I. galbana* was used as a single food source, which was a DMSP-poor and favorite food for *C. sinicus*, and four concentrations of *I. galbana* (10,

25 15, 20, and  $25 \times 10^4$  cells mL-1) were set as concentration contrasts to obtain ingestion and DMS/DMSP information from food concentration. Laboratory studies have confirmed that salinity can significantly influence the ingestion rates of copepod and the S-compounds production of phytoplankton (Tang et al., 1999; Yu et al., 2015). Therefore, four levels of salinities (20, 25, 30, and 35 PSU) were set up to investigate the effects of salinity on copepod grazing and the variations of DMSP and DMS concentrations. Single-factor experiments were performed, and all parameters were run individually.

**2.2.3 IR and CR**

In order to ensure that the copepods were hungry to graze the diets, copepods were starved for 24 h before they were fed with phytoplankton. Copepods were rinsed with sterilized seawater before the beginning of the grazing experiment. Batches of 10 individual copepods in each of 3 replicates were sorted out and transferred to 250 mL polycarbonate bottles. Three

5 bottles containing no grazers were used as controls. All bottles were topped off with suspensions of algae, sealed with parafilm, and fastened onto a spinning plankton wheel (2 rpm) at 15 ± 1 °C in darkness for 24 h. After a 24-h incubation, copepods and fecal pellets were harvested according to the methods of Tang (2001). Water samples were used to analyze cell density, IR, CR, DMS, DMSPp, and DMSPd. The IRs and CRs were calculated according to the equations of Frost (1972). Because no significant differences were found in algae concentrations between the initial and final control bottles, the growth constant (*k*) for algal growth was eliminated from the equations, thus yielding:

$$CR = \frac{V \times \ln(C_1 / C_2^*)}{Nt}$$
(1)

$$IR = CR \times C_1 \tag{2}$$

Where *CR* is the clearance rate (mL ind-1 h-1), *IR* is the ingestion rate (cells ind-1 h-1),  $C_1$  is the initial algal concentration in control bottles (cells mL-1),  $C_2^*$  is the final algal concentration in the experimental bottles (cells mL-1), *t* is

15 the duration of the experiment (h), V and N are the volume (mL) and number of copepods in the experimental bottles (ind), respectively.

Duplicates of 10 mL aliquots of each algal suspension were placed in 40 mL serum bottles that contained 2 mL of 10 mol  $L^{-1}$  KOH solution for DMSP detection. Ten copepods from each bottle were placed individually in serum bottles for DMSPz (DMSP in the copepod body) measurement before gut clearance. Fecal pellets were separated from detritus with a mouth micropipette, rinsed with filtered seawater, and concentrated onto a 47 mm Whatman GF/F filter by gravity filtration to

determine DMSPf (DMSP in the fecal pellets).

20

We ran preliminary experiment to check the effects of the bacteria on DMS concentration. According to the methods of Agostini et al. (2016), the treatment with antibiotics (0.025 g  $L^{-1}$  penicillin G potassium + 0.08 g  $L^{-1}$  streptomycin sulphate + 0.04 g  $L^{-1}$  neomycin sulphate) were used to inhibit the bacteria in the algal culture. When *C. sinicus* were fed on the four

25 diets (*I. galbana*, *C. curvisetus*, *E. huxleyi*, and *Gymnodinium* sp.), no significant differences were found between DMS concentrations in the control (without antibiotics) and those in the treatment (with antibiotics) (data not shown). Therefore, the copepod cultures were not treated with antibiotics in our laboratory experiment to obtain axenicity. Yost and Mitchelmore (2009) reported that antibiotic treatment negatively affected algal growth, which was the other reason for not using antibiotics.

**2.2.4 DMS and DMSP determinations**

According to the methods of Andreae and Barnard (1983), DMS and DMSP concentrations were determined using the purge-and-trap technique by a gas chromatograph, which was equipped with a flame photometric detector (Shimadzu GC-14B). For DMS measurement, 10 mL seawater was directly introduced into a glass purge chamber. Gravity filtering of

5 samples for DMSPd was obtained according to the method of Kiene and Slezak (2006) with minor modifications. The filtrate and unfiltered seawater for DMSPd and total DMSP (DMSPd + DMSPd + DMSPn) measurements were transferred to a 40 mL serum bottle containing 2 mL of 10 mol  $L^{-1}$  KOH solution and kept at 4 °C for at least 24 h to complete cleavage. DMSPd and DMSPp concentrations were determined by the total DMS subtracting the DMS in original seawater. The DMSPf and DMSPz concentrations were calculated by subtracting DMS and DMSPd in filtered water.

**2.3 Statistical analysis 10**

Data were expressed as mean  $\pm$  standard deviation. Student's t test and one-way ANOVA were used to determine the differences between control and treatment samples. Pearson correlations were utilized to assess the relationships between DMS/DMSP and IRs.

**3 Results**

**15 **3.1 Field experiment**

**3.1.1 Abiotic parameter**

The temperature and salinity in Jiaozhou Bay are described in Fig. 2A. The salinity changed from 29.7 PSU to 32.3 PSU, and the lowest and highest salinities occurred in August 2010 (summer) and April 2011 (spring), respectively. The changing range of temperature was 1.72 °C to 25.3 °C, and the lowest and highest temperatures were observed in December 2010 (winter) and August 2010 (summer), respectively.

**3.1.2 Biotic parameters**

Chl a concentrations in surface water ranged from 0.8  $\mu$ g L-1 to 11.0  $\mu$ g L-1, with an average value of 3.1  $\mu$ g L-1. Chl a concentration peaked on April 2011 (spring) (Fig. 2B). The bacterial abundance in surface water ranged from  $5.7 \times 10^7$  cells  $L^{-1}$  to  $1.79 \times 10^8$  cells  $L^{-1}$  (average of 9.68  $\times 10^7$  cells  $L^{-1}$ ), with a peak value in September 2010 (autumn) (Fig. 2B).

25

20

A total of 74 species of zooplankton were identified and were categorized into 5 phyla, 6 classes, 17 orders, 44 families, and 55 genera. Among these species, 31 are Arthropoda, 22 are Coelenterata, 17 are planktonic larvae, 2 are Chaetognatha, 1 are Protozoa, and 1 are Tunicata (Table 2). The species composition of zooplankton varied with months, and the number of species ranged from 19 to 35. The lowest and highest numbers of species occurred in February 2011 (winter) and June 2010

(summer), respectively. In total, 22 copepods were identified, and *Acartia pacifica*, *Acartia bifilosa*, *Centropages abdominalis*, *Eurytemora pacifica*, and *C. sinicus* were the dominant copepods (Table 2 and Fig. 3B).

Temporal variations in the mean zooplankton abundance are shown in Fig. 3A. *Noctiluca scintillans* appeared as the dominant zooplankton, contributing 91% of zooplankton abundance on June 2010. Copepods dominated the zooplankton
community during winter and spring and accounted for 39% to 83% of the total zooplankton (Fig. 3A). The changing trend of copepod abundance showed that the minimum (7.5 individuals m-3) occurred on July 2010, subsequently increased, and peaked (596 individuals m-3) on April 2011. The *C. sinicus* abundance ranged from 0.143 individuals m-3 to 123 individuals m-3, and *C. sinicus* is the dominant copepod from October 2010 to May 2011. The *C. sinicus* abundance showed a similar trend to that of copepod, and *C. sinicus* exhibited the highest and lowest abundances in spring and summer, respectively (Fig. 3B).

**3.1.3 Dilution experiments**

Fig. 4 illustrates the principles of dilution experiments conducted at three stations (C3, D4, and E3) in Jiaozhou Bay during May 2011. At station C3, regression analysis of the dilution series with added nutrients yielded  $\mu = 0.45 \text{ d}^{-1}$ , and  $g = 0.12 \text{ d}^{-1}$ . At station D4, a nearshore at the east of Jiaozhou Bay,  $\mu = 0.49 \text{ d}^{-1}$ , and  $g = 0.11 \text{ d}^{-1}$ . In the present example for station E3,

the Jiaozhou Bay mouth, μ = 0.23 d-1, and g = 1.38 d-1, indicating that mortality in excess of growth (μ < g). Phytoplankton growth rates for the nutrient addition treatments were higher than those for the no-nutrient treatments at the three stations. The results of dilution experiment at three stations (C3, D4 and E3) are presented in Table 3, and μ varied from 0.02 d-1 to 1.29 d-1, with the highest value at station E3 (February 2011). g varied from 0.02 d-1 (June 2010) to 1.38 d-1 (May 2011). Phytoplankton growth rates at coastal stations (D4 and E3) were higher than those at offshore station C3 (except July 2010 and May 2011).

[revised manuscript text omitted]
 DMSPd (r = -0.918, p = 0.001, n = 8). In comparison, DMSPp was observed to be positively correlated with DMSPf (r = 0.491, p = 0.509, n = 4) and DMSPz (r = 0.705, p = 0.295, n = 4).

**4** Discussion**

**4.1 Effects of C. sinicus on DMS/DMSP in the field study**

- 25 In Jiaozhou Bay, C. sinicus was the dominant copepod in 74 species of zooplankton identified from years 2010 to 2011. In this study, no significant correlations were observed between zooplankton, copepod, and/or C. sinicus abundance and DMS/DMSP concentrations in the field study, illustrating that Jiaozhou Bay was a complex ecosystem with different abundances and types of phytoplankton, zooplankton, and/or copepod in the natural environment. Many kinds of copepods inhabited Jiaozhou Bay, and their IRs depended on copepod species, e.g., IRs of *Harpacticus* sp. (Yu et al., 2015) were 10-
- 30 fold of those of *C. sinicus* in this study. The gut contents of *C. sinicus* were checked, and the results showed that the *C. sinicus* preferred to graze on diatom *Chaetoceros curvisetus* and *Thalassiosira nordenskioldi* (data not shown), suggesting that diatom (DMSP-poor algae) were the preferable diet for *C. sinicus*. Zheng et al (2014) and Luo et al (2016) have

investigated the species composition and abundance of phytoplankton in 2010 and 2011 Jiaozhou Bay, respectively. According to their data, the changing trends of species composition, abundance of phytoplankton and dinoflagellate/diatom ratio from June 2010 to May 2011 were presented in Table 4 and Fig. 9. The predominancy of dinoflagellate *Ceratium fusus* was 0.10 in September 2010 (Table 4). The dinoflagellate/diatom ratios in the three months (July, August and September)

- 5 were high among the whole year, and the abundances of dinoflagellate and diatom in September were the highest among the three months (Fig. 9), what is more, the bacterial abundance in September was the highest among the year (Fig. 2). Therefore, the occurrence of high abundances of dinoflagellate and bacteria might were the reason of high DMS and DMSP in September 2010. In February, April and May 2011, the dominant phytoplankton were diatom *Rhizosolenia delicatula*, *Skeletonema costatum*, and *Skeletonema costatum*, and the predominancies were 0.7, 0.99 and 0.68, respectively (Table 4).
- 10 Although the phytoplankton abundances and Chl *a* contents were high during January 2011 to May 2011, the DMSPp and DMSPd concentrations were lower than those in September 2010, suggesting that DMSP concentration not only depend on phytoplankton abundance, but also phytoplankton species and other factors. We evaluated the effects of several agents (i.e., food, diet concentration, and salinity) on DMS and DMSP productions in the laboratory study. Our incubation data showed that copepod grazing increased DMS production, which was consistent with previous investigations on the effects of
- 15 copepod grazing on DMS production (Dacey and Wakeham, 1986; Yu et al., 2015). Consistent with our field study results, no significant correlations between mesozooplankton abundance and the distribution of DMS or DMSPd were also observed in the Gulf of Maine and St. Lawrence in previous studies (Cantin et al., 1996; Matrai and Keller, 1993). Cantin et al. (1996) concluded that mesozooplankton grazing played a minor role in DMS and DMSPd productions in the Gulf of St. Lawrence. The variable conditions in the natural environment can explain the reason for the minor role of zooplankton on DMS/DMSP production and the inconsistent results of field and incubation studies.

**4.2 Dietary effects on copepod grazing and DMS/DMSP production**

[revised manuscript text omitted]

- Wong, C.S., Wong, S.E., Richardson, W.A., Smith, G.E., Arychuk, M.D., and Page, J.S.: Temporal and spatial distribution of dimethylsulfide in the subarctic northeast Pacific Ocean: a high-nutrient-low-chlorophyll region, Tellus B, 57, 317–331, https://doi.org/10.1111/j.1600-0889.2005.00156.x, 2005.
- Yang, G., Li, C., and Sun, J.: Influence of salinity and nitrogen content on production of dimethylsulfoniopropionate (DMSP)
- 10 and dimethylsulfide (DMS) by *Skeletonema costatum*, Chin. J. Oceanol. Limn., 29, 378–386, https://doi.org/10.1007/s00343-011-0087-6, 2011.
  - Yost, D.M. and Mitchelmore, C.L.: Dimethylsulfoniopropionate (DMSP) lyase activity in different strains of the symbiotic alga *Symbiodinium microadriaticum*, Mar. Ecol. Prog. Ser., 386, 61–70, https://doi.org/10.3354/meps08031, 2009.
- Yu, J., Tian, J.-Y., and Yang, G.-P.: Effects of *Harpacticus* sp. (Harpacticoida, copepod) grazing on
   dimethylsulfoniopropionate and dimethylsulfide concentrations in seawater, J. Sea Res., 99, 17–25, https://doi.org/10.1016/j.seares.2015.01.004, 2015.
  - Zheng, S., Sun, X.-X., Zhao, Y.-F., and Sun, S.: Annual variation of species composition and abundance distribution of phytoplankton in 2010 in the Jiaozhou Bay, Mar. Sci., 38, 1–6, 2014. (in Chinese with English abstract)
  - Zhong, S.-L., Wang, Y.-P., Gao, S., Cao, Q.-Y., and Zhu, Y.-H.: Distribution patterns of nannofossil gephyrocapsa oceanica in
- 20 surficial sediments of Jiaozhou Bay, southern Shandong Peninsula, China, Acta Palaeontol. Sin., 40, 505–513, 2001.

Table 1. Sizes and DMSP of phytoplankton cells.

| Algae                  | Dimension (µm)   | Biovolume (µm 3 ) | Carbon (pg cell <math>-1</math> ) | DMSP (fmol cell <math>-1</math> ) |
|------------------------|------------------|------------------------------|----------------------------------------------|----------------------------------------------|
| C. curvisetus          | 7.8 × 4.6        | $9.04 	imes 10^1$            | $1.15 \times 10^1$                           | $1.0 	imes 10^{-1}$                          |
| I. galbana             | $6.0 \times 5.8$ | $1.11 \times 10^2$           | $2.05 \times 10^1$                           | $4.0 	imes 10^{0}$                           |
| Gymnodinium sp. | 14.0 × 12.0      | $1.13 \times 10^3$           | $1.53 \times 10^2$                           | $3.6 \times 10^{1}$                          |
| E. huxleyi             | 2.7 × 3.2        | $1.25 \times 10^1$           | $2.94 	imes 10^{0}$                          | $3.0 \times 10^1$                            |

| Phylum             | Species                                               | 2010.06 | 2010.07 | 2010.08 | 2010.09 | 2010.10 | 2010.11 | 2010.12 | 2011.01 | 2011.02 | 2011.04 | 2011.05 |
|--------------------|-------------------------------------------------------|---------|---------|---------|---------|---------|---------|---------|---------|---------|---------|---------|
| Protozoa           | Noctiluca scientillans Kofoid et Swezy                | 1       | 1       | 1       | 1       | 1       | 1       | 1       | 1       | 1       | 1       | 1       |
| Coelenterata       | Ectopleura dumontieri (Van Beneden)                   |         |         |         |         | 1       |         |         | 1       | 1       |         |         |
|                    | Bougainvillia muscus (Allman)                         | 1       |         |         |         |         |         |         |         |         |         |         |
|                    | Sarsia japonica (Nagao)                               | 1       | 1       | 1       |         |         | 1       |         | 1       |         | 1       | 1       |
|                    | Rathkea octopunctata (M.Sars)                         |         |         |         |         |         |         |         | 1       | 1       |         |         |
|                    | Turritopsis nutricula (McCrady)                       |         |         |         |         |         |         |         |         | 1       |         | 1       |
|                    | Euphysora bigelowi Maas                               |         |         |         |         | 1       |         |         |         |         |         | 1       |
|                    | Euphysora knides Huang                                | 1       |         |         | 1       |         |         | 1       |         |         |         |         |
|                    | Zanclea costata (Linne)                               |         |         | 1       |         |         | 1       |         |         |         |         |         |
|                    | Obelia spo                                            | 1       |         | 1       | 1       | 1       | 1       |         |         |         | 1       | 1       |
|                    | Clutia homisphaarica (Linnaeus)                       | î       | 1       | •       | 1       | 1       | 1       | 1       | 1       | 1       | 1       | 1       |
|                    | Evolutional and English Evolution (English Evolution) |         |         |         | 1       | 1       | 1       | 1       | 1       | 1       | 1       | 1       |
|                    | Eucnenoid menoni Kramp                                |         |         |         |         | 1       | 1       | 1       |         |         |         |         |
|                    | Lirene cytonensis Browne                              |         |         |         |         | 1       | 1       |         |         |         |         | 1       |
|                    | Liriope petraphylia (Chamisso et Eysenhardt)          |         |         |         |         |         | 1       |         |         |         |         |         |
|                    | Sugura chengshanense (Ling)                           | 1       |         |         |         |         |         |         |         |         |         |         |
|                    | Lovenella sp.                                         |         |         |         | 1       | 1       |         |         |         |         | 1       |         |
|                    | Malagazzia carolinae (Mayer)                          | 1       | 1       |         |         |         |         |         |         |         |         |         |
|                    | Proboscidaciyla flavicirrata Brandt                   | 1       |         |         |         |         |         |         | 1       | 1       | 1       | 1       |
|                    | Aequorea conica Browne                                |         |         |         |         |         |         |         | 1       |         |         |         |
|                    | Eutima levuka (Agassiz et Mayer)                      |         |         | 1       | 1       |         |         |         |         |         |         |         |
|                    | Muggiaea atlantica Cunningham                         |         |         | 1       | 1       |         |         |         |         |         |         |         |
|                    | Pleurobrachia globosa Moser                           |         |         |         | 1       | 1       |         |         |         |         |         |         |
|                    | Beröe cucumis Fabricius                               |         |         |         |         |         | 1       |         |         |         |         |         |
| Arthropoda         | Penilia avirostris Dana                               |         | 1       |         |         |         |         |         |         |         |         |         |
|                    | Evadne nordmanni Loven                                |         | 1       |         |         |         |         |         |         |         |         |         |
|                    | Calanus sinicus Brodsky                               | 1       | 1       | 1       | 1       | 1       | 1       | 1       | 1       | 1       | 1       | 1       |
|                    | Paracalamis paraus (Claus)                            | 1       | -       | •       | 1       | 1       | 1       |         | 1       |         |         | 1       |
|                    | Functional parties (Claus)                            |         |         |         | 1       | 1       |         |         |         |         |         |         |
|                    | Provide directory of salo                             |         |         |         |         |         |         |         |         |         | 1       | 1       |
|                    | Pseudodiapiomus popiesia Shen                         |         |         |         |         |         |         | 1       | 1       |         |         |         |
|                    | Sino calanus tenellus (Kikuchi)                       |         |         |         |         |         |         | 1       | 1       |         |         |         |
|                    | Centropages abdominalis Sato                          | 1       | 1       |         |         |         | 1       | 1       | 1       | 1       | 1       | 1       |
|                    | Centropages dorsispinatus Thompson et Scott           |         |         | 1       | 1       | 1       | 1       |         |         |         |         |         |
|                    | Centropages tenuiremis Thompson et Scott              | 1       | 1       | 1       | 1       | 1       |         |         |         |         |         | 1       |
|                    | Calanopia thompsoni A.Scott                           | 1       | 1       | 1       | 1       | 1       | 1       |         |         |         |         |         |
|                    | Labidocera pavo (Dana)                                |         |         | 1       |         |         |         |         |         |         |         |         |
|                    | Labidocera euchaeta Giesbrecht                        |         |         |         | 1       |         | 1       | 1       | 1       | 1       |         |         |
|                    | Labidocera bipinnata Tanaka                    | 1       | 1       | 1       | 1       | 1       | 1       |         |         |         |         | 1       |
|                    | Pontellopsis tenuicauda (Giesbrecht)                  | 1       |         |         |         |         |         |         |         |         |         |         |
|                    | Acartia bifilosa Giesbrecht                           | 1       |         |         |         | 1       | 1       | 1       | 1       | 1       | 1       | 1       |
|                    | Acartia pacifica Steuer                               | 1       | 1       | 1       | 1       | 1       | 1       |         |         |         |         | 1       |
|                    | Tortanus destrilobatus Chen and Zhang                 | 1       | 1       | 1       | 1       | 1       |         |         |         |         | 1       | 1       |
|                    | Tortanus spinicaudatus Shen et Bai                    |         |         | 1       | 1       |         |         |         |         |         |         |         |
|                    | Tortanus farcinatus (Giesbrecht)                      |         |         |         |         | 1       | 1       |         |         |         |         |         |
|                    | Oithong similis Claus                                 |         |         |         |         | 1       | •       |         | 1       |         | 1       | 1       |
|                    | Convegous affinis Momumichi                           |         | 1       |         | 1       |         |         |         |         | 1       | 1       |         |
|                    | Corycaeus ajjurus Memurrieni                          |         | 1       | 1       | 1       |         | 1       |         | 1       | 1       |         |         |
|                    | Harpacticoida                                         |         |         |         |         |         | 1       | 1       | 1       | 1       |         |         |
|                    | Monstrilla sp.                                        |         |         |         | 1       |         | 1       |         |         |         |         |         |
|                    | Themisto gracilipes (Norman)                          |         |         |         |         |         |         |         |         |         | 1       | 1       |
|                    | Acanthomysis longirostris 👔                           |         |         |         |         | 1       | 1       | 1       | 1       | 1       | 1       | 1       |
|                    | Acetes japonicus Kishinouye                           | 1       |         |         |         |         |         |         |         |         |         |         |
|                    | Gammaridea                                            | 1       |         | 1       | 1       |         | 1       | 1       |         |         | 1       | 1       |
|                    | Caprella sp.                                          |         |         | 1       | 1       |         |         |         |         |         |         | 1       |
|                    | Microniscus sp.                                       |         |         |         | 1       | 1       |         |         |         |         |         |         |
|                    | Leueon sp.                                            |         |         |         |         |         |         | 1       | 1       | 1       |         |         |
| Chaetognatha       | Sagitta nagae Alvarino                                |         |         | 1       |         | 1       | 1       | 1       |         |         |         | 1       |
|                    | Sagitta crassa Tokioka                                | 1       | 1       | 1       | 1       | 1       | 1       | 1       | 1       | 1       | 1       | 1       |
| Tunicata           | Oikoplawa dioica Fol                                  | -       |         | •       | -       | 1       | 1       |         |         |         |         | 1       |
| Displatonia logues | Trashashara lama                                      |         |         |         |         |         | •       |         |         |         |         |         |
| Planktonic laivae  |                                                       | 1       |         |         |         |         |         |         |         |         |         |         |
|                    | Polychaeta larva                                      | 1       |         | 1       | 1       | 1       | 1       | 1       | 1       | 1       |         | 1       |
|                    | Gastropod post larva                                  | 1       | 1       | 1       | 1       | 1       |         |         |         |         | 1       | 1       |
|                    | Bivalve larva                                         | 1       |         | 1       |         | 1       | 1       |         | 1       | 1       |         | 1       |
|                    | Nauplius larva (Copepoda)                             |         |         |         |         | 1       |         |         |         |         | 1       |         |
|                    | Nauplius larva (Cirripedia)                           | 1       |         |         |         |         |         |         |         |         | 1       |         |
|                    | Cypris larva                                          | 1       |         |         |         |         |         |         |         |         |         |         |
|                    | Macrura larva                                         | 1       | 1       | 1       | 1       | 1       |         |         |         |         | 1       | 1       |
|                    | Brachyura zoea larva                                  | 1       | 1       | 1       | 1       | 1       | 1       |         |         |         | 1       | 1       |
|                    | Megalopa larva                                        | 1       | 1       | 1       | 1       |         |         |         |         |         |         |         |
|                    | Porcellana zoea larva                                 | -       | 1       | 1       | 1       |         |         |         |         |         |         |         |
|                    | Alima larva                                           | 1       | 1       | -       | *       |         |         |         |         |         |         |         |
|                    | Onbionluteus larza                                    | •       | •       | •       | 1       |         |         |         |         |         |         |         |
|                    | Controputeus farva                                    |         |         |         |         |         |         |         |         |         |         |         |
|                    | Echnopruteus farva                                    |         |         | T       | 1       | T       | 1       | 1       | 1       | 1       |         |         |
|                    | Echinodermata larva                                   |         |         |         | 1       |         | 1       |         |         |         |         |         |
|                    | Fish eggs                                             | 1       | 1       |         |         |         |         |         |         |         | 1       | 1       |
|                    | Fish larva                                            | 1       | 1       | 1       |         |         |         |         | 1       | 1       | 1       | 1       |

**Table 2. Species of zooplankton in Jiaozhou Bay.**

"1" means appearance.

| Sampling time | Station | $\mu$ (d -1 ) | $g(d^{-1})$ | $r^2$ |
|---------------|---------|--------------------------|-------------|-------|
| 2010.06       | C3      | 0.31                     | 0.26        | 0.52  |
|               | D4      | 0.44                     | 0.16        | 0.20  |
|               | E3      | 0.68                     | 0.65        | 0.57  |
| 2010.07       | C3      | 0.36                     | 0.02        | 0.15  |
|               | D4      | 0.33                     | 0.55        | 0.46  |
|               | E3      | 0.66                     | 0.33        | 0.65  |
| 2010.08       | C3      | 0.38                     | 0.27        | 0.59  |
|               | D4      | 0.61                     | 0.60        | 0.76  |
|               | E3      | 0.71                     | 0.45        | 0.48  |
| 2010.09       | C3      | 0.27                     | 0.40        | 0.10  |
|               | D4      | 0.27                     | 0.10        | 0.20  |
|               | E3      | 0.28                     | 0.59        | 0.40  |
| 2010.10       | C3      | 0.85                     | 0.22        | 0.47  |
|               | D4      | 0.88                     | 0.44        | 0.39  |
|               | E3      | 0.93                     | 0.96        | 0.28  |
| 2010.11       | C3      | 0.12                     | 0.31        | 0.54  |
|               | D4      | 0.12                     | 0.43        | 0.42  |
|               | E3      | 0.49                     | 0.29        | 0.43  |
| 2010.12       | C3      | 0.13                     | 0.37        | 0.61  |
|               | D4      | 0.96                     | 0.15        | 0.25  |
|               | E3      | 0.36                     | 0.30        | 0.71  |
| 2011.01       | C3      | 0.06                     | 0.28        | 0.60  |
|               | D4      | 0.13                     | 0.33        | 0.23  |
|               | E3      | 0.45                     | 0.13        | 0.20  |
| 2011.02       | C3      | 0.02                     | 0.49        | 0.25  |
|               | D4      | 1.12                     | 1.15        | 0.65  |
|               | E3      | 1.29                     | 0.67        | 0.40  |
| 2011.04       | C3      | 0.25                     | 0.48        | 0.41  |
|               | D4      | 0.52                     | 0.94        | 0.52  |
|               | E3      | 0.48                     | 0.66        | 0.70  |
| 2011.05       | C3      | 0.45                     | 0.12        | 0.39  |
|               | D4      | 0.49                     | 0.11        | 0.52  |
|               | E3      | 0.23                     | 1.38        | 0.66  |

Table 3. Growth and grazing rates of chlorophyll *a* estimated from the dilution technique.

**Table** 4. Dominant species and the predominancy of phytoplankton in the Jiaozhou Bay from June 2010 to May 2011 (citedfrom Zheng et al. (2014) and Luo et al. (2016)).

| Dominant species             | Predominancy |         |         |         |         |         |         |         |         |         |         |         |
|------------------------------|--------------|---------|---------|---------|---------|---------|---------|---------|---------|---------|---------|---------|
|                              | 2010.06      | 2010.07 | 2010.08 | 2010.09 | 2010.10 | 2010.11 | 2010.12 | 2011.01 | 2011.02 | 2011.03 | 2011.04 | 2011.05 |
| Coscinodiscus asteromphalus  |              | 0.04    | 0.23    | 0.13    |         | 0.1     | 0.03    |         |         |         |         |         |
| C. wailesii                  |              | 0.04    | 0.03    |         |         |         |         |         |         |         |         |         |
| Coscinodiscus spp.           |              |         | 0.03    | 0.06    |         | 0.03    |         | 0.07    |         | 0.13    |         |         |
| Actinocyclus ehrenbergii     |              |         | 0.03    |         |         | 0.12    | 0.14    |         |         |         |         |         |
| Lauderia annulata            | 0.16         |         |         |         |         |         |         |         |         |         |         |         |
| Skeletonema costatum         |              |         |         | 0.17    | 0.33    | 0.03    |         |         |         |         |         |         |
| Leptocylindrus danicus       |              |         |         |         |         |         |         | 0.21    | 0.04    | 0.04    |         |         |
| Rhizosolenia delicatula      |              |         |         |         |         | 0.07    |         |         | 0.7     |         |         |         |
| R. imbricata                 |              |         |         |         |         |         | 0.13    | 0.02    |         |         |         |         |
| R. stolterfothii             | 0.04         |         |         |         |         |         | 0.04    | 0.11    |         |         |         |         |
| R. alata f. indica           |              |         |         |         |         |         |         |         |         | 0.02    |         |         |
| Guinardia flaccida           |              |         |         |         |         |         |         | 0.1     |         |         |         |         |
| Schroederella delicatula     |              |         |         |         |         |         |         | 0.05    |         |         |         |         |
| Thalassiosira nordenskioldii |              |         |         |         |         |         |         |         |         | 0.07    |         |         |
| T. pacifica                  |              |         |         |         |         |         |         |         |         | 0.02    |         |         |
| Chaetoceros debilis          |              |         |         |         |         |         |         | 0.02    |         |         |         |         |
| C. didymus                   |              |         |         |         |         |         |         |         |         |         |         | 0.02    |
| C. densus                    | 0.45         |         |         |         |         | 0.06    | 0.14    |         |         |         |         |         |
| C. curvisetus                |              |         |         |         | 0.05    |         |         | 0.1     |         |         |         |         |
| C. lorenzianus               |              |         |         | 0.05    |         |         |         |         |         |         |         |         |
| C. pseudocurvisetus          |              |         | 0.05    | 0.02    |         |         |         |         |         |         |         |         |
| C. teres                     |              |         |         |         |         | 0.04    |         |         |         |         |         |         |
| Chaetoceros spp.             |              |         |         |         |         |         |         | 0.02    |         |         |         |         |
| Biddulphia sinensis          |              |         | 0.06    |         |         | 0.02    |         |         |         |         |         |         |
| Eucampia zoodiacus           |              | 0.07    | 0.05    | 0.03    |         |         |         | 0.02    |         |         |         |         |
| Skeletonema costatum         |              |         |         |         |         |         |         |         |         |         | 0.99    | 0.68    |
| Asterionella. Kariana        |              |         |         |         |         |         |         | 0.02    | 0.02    |         |         |         |
| Ditylum brightwelii          |              |         |         |         | 0.02    |         |         |         |         |         |         |         |
| Cerataulina bergonii         |              |         | 0.06    |         | 0.06    |         |         |         |         |         |         |         |
| Navicula membranacea         |              |         |         |         |         | 0.05    | 0.02    | 0.03    |         |         |         |         |
| Pseudonitzschia pungens      | 0.18         |         |         |         | 0.05    |         |         |         |         |         |         |         |
| Nitzschia paradoxa           |              | 0.02    |         |         | 0.06    | 0.17    | 0.05    |         |         |         |         |         |
| Protoperidinium depressum    |              | 0.02    | 0.02    |         |         |         |         |         |         |         |         |         |
| Ceratium fusus               |              | 0.12    | 0.04    | 0.1     | 0.03    |         |         |         | 0.02    |         |         |         |
| C. macroceros                |              | 0.03    |         |         |         |         |         |         |         |         |         |         |
| C. tripos                    |              | 0.05    | 0.02    |         |         |         |         |         |         |         |         |         |
| Noctiluca scintillans        | 0.03         | 0.21    |         |         |         |         |         |         |         | 0.04    |         |         |

(Note: blank area means the predominancy

Figure 1. Location of fieldwork sampling stations in Jiaozhou Bay.

---

## Author Comment (AC3) · 12 Jun 2018

**Author's response for Referee #2:**

The manuscript "Role of *Calanus sinicus* (Copepoda, Calanoida) on dimethylsulfide production in Jiaozhou Bay" by Juan et al attempts to understand the role of copepod *Calanus sinicus* in DMS production in Jiaozhou Bay through insitu observations and lab experiments. The authors followed a yearly cycle of insitu observations on temperature, salinity, Chl *a*, TBC, zooplankton enumeration and speciation, DMS, DMSPp and DMSPd at 10 stations in the Jiaozhou Bay. They also performed lab experiments wherein they conducted zooplankton grazing experiments on select phytoplankton species to see the impact on DMS production. Though the hard work put in by the authors in commendable, there is a major disconnect between observations and lab experiments. Their observation on DMSP transfer from phytoplankton to copepod body, fecal pellet to seawater is not new and has been proposed quite some time back (Tang et al; Belviso et al). The authors mention that data from the field measurements showed that *Calanus sinicus* did not have any apparent effect on DMS/DMSP production and then the authors go ahead to perform a complex grazing experiment to see the impact of grazing on DMS production. If the field data did not show any connect, what was the aim to perform the lab experiments? Perhaps the authors should have checked the gut content to see what species the copepod preferred to feed on? This might have given some clues on how to proceed. Also, the insitu observation part is not clear. Some of the concerns with regards to this work is jotted below.

Response: The dilution experiment have been added in the revised submisstion, and the sentence of "Data from the field experiment showed that *C. sinicus* has no apparent effect on DMS/DMSP production" have been deleted .

   The gut contents of *Calanus sinicus* were checked, the results showed that the *C. sinicus* preferred to feed on diatom *Chaetoceros curvisetus* and *Thalassiosira nordenskioldi*, suggesting that diatom (DMSP-poor algae) were the preferable diet for *C. sinicus*, which has been added in 4.1. Dilution

experiments of three stations had been done on the shipboard, and the supplement details has been added in the revised manuscript according to the two referees' suggestions.

1. The authors mention time series sampling at 10 locations, but figure 2 shows data for only one site, which is this site? Or is this averaged data? If averaged then include standard deviation.

Response: Figure 2 shows data for the average data of 10 locations, and the standard deviation were added in the Figure 2.

2. What is the reason for the increase in DMSPp&d (and marginal increase in DMS) during September 2010? What is the major phytoplankton species during spring? As this might have answered the high DMSP observed during that time.

Response: September is in the season of autumn (not spring) in Jiaozhou Bay. Zheng et al (2014) and Luo et al (2016) investigated the species composition and abundance of phytoplankton in 2010 and 2011 in Jiaozhou Bay (Table 4), respectively. From their results, we found that the major phytoplankton species in September 2010 were diatom *Skeletonema costatum* and *Coscinodiscus asteromphalus*, and dinoflagellate *Ceratium fusus*, with the predominancy of 0.17, 0.13 and 0.10, respectively (Table 4). The dinoflagellate/diatom ratio in September is about 0.2 (Fig. 9). On the other hand, the mean bacteria abundance in September is the highest in the year (Fig. 2). Therefore, the dinoflagellate and bacteria might explain the increase in $DMSP_p$ and $DMSP_d$ (and marginal increase in DMS) during September 2010. These were added in the part 4.1.

3. In terms of copepod, the authors mention *Calanus sinicus* as the predominant copepod, but that does not seem to be the case as *Eurytemra pacific* was also dominant during three sampling with April 2011 showing maximum abundance.

Response: There is a mistake for the use of the word "predominant", so "predominant" has been

replaced with "dominant". See Page 7 Line 2.

4. In the feed (diet) experiment it is clear that the copepods prefer *I. galbana* and *C. curvisetus* compared to *E. Huxleyi* and *Gymnodinium* sp. There is not much difference in DMS production in the treatment when compared to the control. On the contrary DMS production dropped in the case of *E. Huxleyi* and *Gymnodinium* sp. in comparison to *I. galbana* and *C. curvisetus* which showed marginal increase in DMS production.

Response: Yes, the copepods prefer *I. galbana* and *C. curvisetus* compared to *E. Huxleyi* and *Gymnodinium* sp. Although there is not much difference in DMS production in the treatment when compared to the control, but we have repeated several times and found that DMS production increased in the case of copepod grazing on *I. galbana* and *C. curvisetus*, on the contrary DMS production dropped in the case of *E. Huxleyi* and *Gymnodinium* sp. The cellular DMSP concentration in algae might be the reason.

5. Line 26: ...in May 2011) and had no apparent....

Response: "and has no apparent effect on DMS/DMSP production" has been deleted. The sentence were changed into "Data from the field experiment showed that *C. sinicus* was the dominant copepod in Jiaozhou Bay (up to 123 individuals $m^{-3}$ in May 2011) and preferred to graze on diatom. DMS and DMSP concentrations not only depend on phytoplankton abundance, but also phytoplankton species and other factors.", see Page 1 Line 16-18.

6. Introduction: Line 37-38: The authors mention that there was close scrutiny on DMSP, what kind of scrutiny, please elaborate.

Response: "recently came under close scrutiny" has been removed according to the comments of referee #1. See Page 1 Line 27.

7.  Line 48: what kind of biotic and abiotic factors? Elaborate.

Response: "Numerous abiotic and biotic factors" has been changed into "Numerous biotic factors (i.e., phytoplankton, zooplankton, bacteria, and virus) and abiotic factors (i.e., temperature, salinity, light, and nutrient)". See Page 2 Line 4-5.

8.  Line 49: replace 'account' with 'consume'.

Response: 'account' has been replaced with 'consume'. See Page 2 Line 6.

9.  Line 51: the authors mentioned that role of zooplankton grazing on DMSP biogeochemical processes are scarce. And later in the same paragraph include detailed studies on the impact of zooplankton grazing on DMS production. The aim for carrying out the grazing work needs to clearly spelt out.

Response: There were inconsistency about evaluating zooplankton grazing in the paragraph, therefore, the sentence "and knowledge about the role of zooplankton grazing in the DMSP biogeochemical processes is scarce" has been deleted. See Page 2 Line 7.

10.  Line 62: delete 'Brodsky'.

Response: 'Brodsky' has been deleted. See Page 2 Line 15.

11.  Line 65: what field experiments were performed? Are the authors referring to field measurements?

Response: Yes, field experiments were referred to field measurements, and the sentence has been replaced with "field measurements and laboratory experiments". See Page 2 Line 17.

**Materials and methods:**

12.  The authors mention collection of samples from 10 stations. ① I assume these are surface samples? How were the samples collected? Niskin sampler or any other sampler? ② Zooplankton samples were collected by vertical tows, mention depth or range of depth from where the vertical tows were done.

Response: ① Yes, the water samples were the surface samples, and the samples were collected by a Niskin sampler, see Page 3 Line 30. ② Zooplankton samples were collected by vertical tows from the bottom to the surface, the depth varied from 3 m to 28 m according to different station. See Page 3 Line 2.

13. Line 91: spelling correction 'Whatman' GF/F filters. There are many grammatical errors in the manuscript. Only a few are pointed out. The authors need to correct these.

Response: 'Waterman GF/F filters' has been replaced with 'Whatman GF/F filters'. See Page 3 Line 30. We have checked the grammars in the manuscript.

14. Line 334: Did the authors measure acrylic acid as a deterrent against grazing?

Response: No, we did not measure acrylic acid, so 'acrylic acid or' had been deleted. See Page 11 Line 30.

15. Line 345-346: 'In this study, .......which in turn reduced DMS/DMSP production'. This dose not seem to be the case as DMS production was high in *Gymnodinium* sp. and *E. Huxleyi* as seen from control and did not depend on grazing.

Response: The comments of the referee is right, and the sentence 'In this study, .......which in turn reduced DMS/DMSP production' has been deleted. See Page 12 Line 6.

16. Figure 1: Include latitude and longitude or specify 'N' and 'E'.

Response: 'N' and 'E' were added in the Figure 1.

17. Figure 3: (B) may be deleted as species wise in shown in (C).

Response: Figure 3 (B) has been deleted.

18. One of the important parameters that this work lacks is phytoplankton speciation of the natural samples. In the absence of that data, understanding DMSP variation becomes difficult.

Response: Zheng et al (2014) and Luo et al (2016) have investigated the species composition and abundance of phytoplankton in 2010 and 2011 Jiaozhou Bay, respectively. According to their data, the changing trend of species composition, abundance of phytoplankton and dinoflagellate/diatom ratio from June 2010 to May 2011 were presented in Table 4 and Fig. 9. The predominancy of dinoflagellate *Ceratium fusus* was 0.10 in September 2010 (Table 4). The dinoflagellate/diatom ratios in the three months (July, August and September) were high among the whole year, and the abundances of dinoflagellate and diatom in September were the highest among the three months (Fig. 9), what is more, the bacterial abundance in September was the highest among the year (Fig. 2). Therefore, the occurrence of high abundances of dinoflagellate and bacteria might were the reason of high DMS and DMSP in September 2010. In February, April and May 2011, the dominant phytoplankton were diatom *Rhizosolenia delicatula*, *Skeletonema costatum* and *Skeletonema costatum,* and predominancies were 0.7, 0.99 and 0.68, respectively (Table 4). Although the phytoplankton abundances and Chl *a* contents were high during January 2011 to May 2011, the $DMSP_p$ and $DMSP_d$ concentrations were lower than those in September 2010, suggesting that DMSP concentration not only depend on phytoplankton abundance, but also phytoplankton species and other factors. See 4.1, Table 4, and Fig. 9.

19. Grazing by zooplankton on phytoplankton is an important part which results in DMSP going from particulate (within cell) to dissolved (outside) and further the action by DMSP lyase (both by phytoplankton as well bacterial lysis) results in high DMS production. Grazing is studied by looking at the gut content or by isotopic work, neither being dong in the present study, it's difficult to address DMS production to grazing.

Response: Gut content in the copepod were checked in the field study, and found that *C. sinicus* preferred to graze on diatom *Chaetoceros curvisetus* and *Thalassiosira nordenskioldi* (data not shown),

suggesting that diatom (DMSP-poor algae) were the preferable diet for the copepod *C. sinicus*. See Page 10 Lines 30-32. Isotopic work will be done in the future study to address DMS production to grazing.

20. And finally, there is a complete disconnect between field results and the basis for performing grazing experiments in the laboratory.

Response: Dilution experiments of three stations had been done on the shipboard from June 2010 to May 2011, and the data have been added in the revised manuscript according to the two referees' suggestion. We did not put these data in the previous submission because that copepod numbers were usually low ($< 1$ $L^{-1}$) compared with microzooplankton. See 2.1.3, 3.1.3, Table 3, Fig. 4. Then, the grazing experiments were investigated both in field and in laboratory.